# Recurrent mutations in the stress regulator Cap1 reveal a trade-off between azole resistance and oxidative stress response in *Candida albicans*

Xin Zhou[1], Audrey Hilk[1], Norma V. Solis[2], Nancy Scott[1], Christopher Zajac[1], Scott G. Filler[2,3], Anna Selmecki[1]*

1 Department of Microbiology and Immunology, University of Minnesota, Minneapolis, Minnesota, United States of America, 2 Division of Infectious Diseases, Lundquist Institute for Biomedical Innovation at Harbor UCLA Medical Center, Torrance, California, United States of America, 3 David Geffen School of Medicine at UCLA, Los Angeles, California, United States of America

* selmecki@umn.edu

## Abstract

Drug resistance is a critical challenge in treating life-threatening fungal infections. Here, we uncover a mechanism of acquired azole resistance in *Candida albicans* through mutations in *CAP1*, encoding a conserved fungal transcription factor that mediates the oxidative stress response. We analyzed 300 clinical isolates and identified 25 distinct *CAP1* missense or nonsense mutations, with many occurring within the DNA-binding domain. We identified two nearly identical *CAP1* heterozygous nonsense mutations, one in an isolate obtained from a bloodstream infection and one in a population of cells undergoing adaptation to fluconazole *in vitro*. Both *CAP1* nonsense mutations resulted in loss of the C-terminal nuclear export signal, leading to nuclear retention of Cap1 and subsequent activation of genes associated with the oxidative stress response and drug transport. The *CAP1* C-terminal truncations conferred significant fitness advantages in the presence of fluconazole, both *in vitro* and in a murine model of candidiasis. Strikingly, we discovered a therapeutic vulnerability: azole concentrations above the minimal inhibitory concentration were fungicidal to mutants with the *CAP1* C-terminal truncation. The fungicidal effect was attributed to both elevated azole-induced reactive oxygen species and a compromised oxidative stress response in Cap1-truncated cells. Our results provide novel characterization of *de novo CAP1* point mutations emerging in both laboratory and clinical contexts, elucidate the mechanisms underlying Cap1-regulated stress responses, and reveal a potential therapeutic target for overcoming drug resistance in *C. albicans* infections.

## Introduction

*Candida albicans* is the most common opportunistic fungal pathogen in humans. In the United States alone, it causes ~46,000 invasive infections each year, mostly

**Data availability statement:** All whole genome sequences and RNA sequences are available in the NCBI Sequence Read Archive repositories BioProject accession numbers PRJNA1299409 and PRJNA1299481. All flow cytometry raw data (.fcs files) and gates are available in https://doi.org/10.5281/zenodo.18250101. All computational scripts used to generate figures and perform whole-genome sequence alignment and variant calling are publicly available through the Selmecki Lab GitHub page (https://github.com/selmeckilab) and Zenodo (https://doi.org/10.5281/zenodo.18250101). All raw numerical data (.xlsx files) used to generate figures are available in the Supporting Information and in https://doi.org/10.5281/zenodo.18250101.

**Funding:** Funding for this work was supported in part by the National Institute of Allergy and Infectious Diseases (R01-AI143689), National Science Foundation (DBI-2320251), and the Burroughs Wellcome Fund Investigator in the Pathogenesis of Infectious Diseases Award (#1020388) to AS. The funders had no role in study design, data collection and analysis, decision to publish, or preparation of the manuscript.

**Competing interests:** The authors have declared that no competing interests exist.

affecting people with weakened immune systems [1–5]. Even with current antifungal therapies, invasive infections caused by *C. albicans* can be deadly, with mortality rates reaching up to 60%, highlighting a significant medical concern [1–3,6]. During infection, *C. albicans* is exposed to diverse stressors, including oxidative stress from the host's immune cells and the effects of antifungal drug exposure [7]. Treatment options for invasive candidiasis are limited to just three main classes of antifungal drugs: azoles, echinocandins, and polyenes [4]. Azoles, such as fluconazole (FLC), are widely used due to their oral bioavailability and minimal side effects; however, most azoles are fungistatic rather than fungicidal [8,9]. Recent studies suggest that, in addition to inhibiting ergosterol biosynthesis, azoles can exert fungicidal effects by disrupting cell wall integrity or activating the reactive oxygen species (ROS)-dependent programmed cell death response [10,11].

*C. albicans* has regulatory machinery that coordinates a cellular defense against both oxidative and antifungal drug stress. Central to this regulatory network is the basic leucine zipper (bZip) domain transcription factor Cap1, which functions as a molecular bridge between the oxidative stress response (OSR) and azole resistance across diverse yeast species [12–19]. The *C. albicans CAP1* gene was initially identified as an FLC-resistance gene when heterologous expression of a truncated *CAP1* allele (containing nonsense mutation P334*) in *Saccharomyces cerevisiae* led to increased expression of the multidrug efflux pump *MDR1* and increased FLC resistance [12,20,21]. Additionally, *CAP1* regulates antioxidant processes that contribute to resistance against phagocyte-mediated killing and lipid peroxidation [22–25]. Despite these established roles, the clinical relevance of *CAP1* mutations remains poorly defined.

*C. albicans* Cap1 shares structural and functional homology with *S. cerevisiae* Yap1 and has conserved orthologs across eukaryotes, including *Schizosaccharomyces pombe* (Pap1), *Candida glabrata* [*Nakaseomyces glabratus*] (CgAp1), *Aspergillus fumigatus* (Yap1), *Kluyveromyces lactis* (Yap1), and *Arabidopsis thaliana* (Yap1) [18,19,26–30]. The most highly conserved features across these orthologs are the cysteine-rich domains (CRDs) positioned at both the N-terminus and C-terminus, with the latter containing CSE repeats (Cysteine, Serine, Glutamic acid) [12–17,30]. The cysteine-rich domains mediate a conserved regulatory mechanism whereby oxidative stress triggers intramolecular disulfide bond formation between N- and C-terminal cysteines (oxidized Cap1), masking the nuclear export signal within the C-terminal CRD (C-CRD) and promoting nuclear retention for transcriptional activation [31,32]. Consistent with this regulatory model, C-terminal disruption via missense mutations in CSE repeats (e.g., CSE477) or C-CRD deletions constitutively activates Cap1/Yap1 gene targets and confers enhanced resistance to azoles and oxidative stress in both *C. albicans* and *A. fumigatus* [12,15,21,30,33]. However, despite the demonstrated capacity of C-CRD mutations to drive resistance in experimental settings and the critical importance of this domain in Cap1 regulation, no naturally occurring mutations within this region have been identified in clinical isolates, leaving in question whether C-terminal truncations can emerge under therapeutic pressure and

how such mutations might simultaneously affect both drug resistance and oxidative stress responses in clinically relevant contexts.

Transcriptional activation of oxidized *CAP1* or C-terminally disrupted Cap1 drives both self-upregulation and enhanced expression of downstream targets, leading to azole resistance and other adaptive phenotypes [12,15,21,25,30,33]. However, C-terminal truncation of Cap1 does not substantially alter its ability to bind downstream targets, as 60 out of the total 89 targets were common to both the wild-type and a mutant with truncated Cap1 by ChIP-chip [15]. These 60 common targets are enriched for genes involved in the oxidative stress response (*CAP1*, *SOD1*, *CAT1*), drug stress response (*PDR16*, *MDR1*, *FLU1*), and other cellular functions such as phospholipid transport and nitrogen utilization [15]. The Cap1 transcriptional program becomes particularly relevant given that azole antifungals themselves induce endogenous ROS in diverse fungal species [34–37]. Notably, generation of ROS is most pronounced with 'fungicidal azoles' like miconazole and itraconazole, whereas the results with the fungistatic azole fluconazole have been inconsistent. How azole-induced ROS influence *CAP1* transcriptional regulation and whether this impacts drug resistance and oxidative stress responses remains poorly understood—a critical gap, given that fungal cells may navigate both antifungal pressure and host-derived oxidative stress during infection [34–38].

To investigate the molecular mechanisms of Cap1-mediated drug resistance in *C. albicans*, we compared the physiological effects of different Cap1 variants identified across 300 clinical isolates and experimentally evolved isolates. We identified two nearly identical *CAP1* heterozygous nonsense mutations, one in an isolate obtained from a bloodstream infection and one from cells undergoing adaptation to FLC *in vitro*. Both *CAP1* nonsense mutations resulted in loss of the C-terminal nuclear export signal, leading to nuclear retention of Cap1 and subsequent activation of genes associated with oxidative stress response and drug transport. Paradoxically, upon FLC treatment, the Cap1 truncation mutants increase endogenous ROS and simultaneously decrease expression of genes involved in the oxidative stress response, effectively converting the azole drug from fungistatic to fungicidal at concentrations above the MIC (minimum inhibitory concentration). Despite this paradox, FLC treatment can significantly increase the virulence of mutants with the Cap1 truncation in a murine model. These findings reveal a previously undescribed trade-off between drug resistance and oxidative stress tolerance associated with *C. albicans* Cap1.

## Results

### Heterozygous *CAP1* mutation evolves rapidly *in vitro* and confers a significant fitness advantage in fluconazole

We determined the frequency and dynamics of both large and small genomic changes in *C. albicans* isolates during adaptation to different concentrations of FLC *in vitro* [39]. In one *in vitro* evolution experiment, we identified a *de novo* heterozygous nonsense mutation in *CAP1* (*CAP1/CAP1^C446*^*) after 10 passages (P0-P10) in the presence of a low, fungistatic concentration of FLC (1 μg/ml FLC), which correlated with reduced FLC susceptibility (Figs 1A and 1B). To understand the evolutionary trajectory of the *CAP1* mutation over the course of the experiment, we performed phenotypic and whole-genome sequence (WGS) analyses at the population and single-colony level. The population at P1 and all subsequent populations had an increased growth rate (area under the curve, AUC) in the presence of ≤ 2 μg/ml FLC compared to the progenitor population at P0 (Fig 1A). The allele frequency of the *CAP1* nonsense mutation increased rapidly from 8.1% at P1 to 57.9% at P2 and remained at ~50% for all subsequent populations (Figs 1B and S1 Fig). At P1, 9 out of 94 single colonies (9.6%) had increased growth rate in FLC, and the heterozygous *CAP1* mutation was found only in the resistant colonies by WGS (S1 Fig and S1 Data). At P10, all single colonies were heterozygous for *CAP1/CAP1^C446*^* by WGS and had a ~2-fold increased fitness in 1 μg/ml FLC when compared to the progenitor in a head-to-head assay (Fig 1C). We did not observe recurrent point mutations or chromosome copy number changes other than the *CAP1* mutation. In summary, a heterozygous nonsense mutation in *CAP1* evolved rapidly *in vitro* and conferred a high fitness advantage in 1–2 μg/ml FLC.

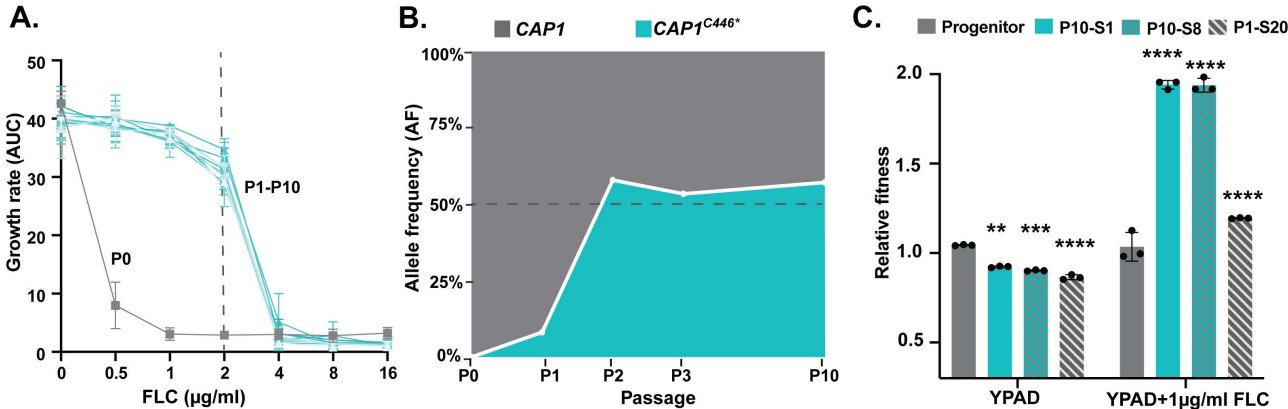

**Fig 1. Heterozygous *CAP1* mutation evolves rapidly and confers a significant fitness advantage in the presence of fluconazole. A.** Growth rate (area under the curve, AUC, Y-axis) of *in-vitro* evolved populations from passage 0 to passage 10 (P0-P10) in the presence of different concentrations of FLC (0-16 µg/ml FLC, X-axis). Gray: P0, and P1-P10: turquoise. The dashed line indicates minimum concentration that inhibits 50% growth of the P1-P10 population ($MIC_{50}$). **B.** Allele frequency (Y-axis) of wild-type *CAP1/CAP1* (gray) and *CAP1/CAP1^{C446*}* (turquoise) quantified from whole-genome sequence analysis at P0, P1, P2, P3 and P10 (X-axis). 50% allele frequency, expected for a heterozygous mutation, is indicated with a dashed line. Additional whole-genome sequence analysis provided in S1 Fig. **C.** Relative fitness of single colonies with or without the *CAP1* mutation, P10-S1 and P10-S8 (*CAP1/CAP1^{C446*}*), and P1-S20 (*CAP1/CAP1*) with the progenitor as the control. Relative fitness determined using a head-to-head competition with the wild-type control (methods). Values are mean±SD calculated from three biological replicates. Data were assessed for normality with a Shapiro–Wilk test, and significant differences between P10-S1/P10-S8/P1-S20 and the progenitor across different environments were calculated using two-way ANOVA with Šidák multiple comparisons test (two-sided); ** $P<0.01$, *** $P<0.001$, ****$P<0.0001$; the exact *P* values are **0.0011, ***0.0002 and ****$<0.0001$ for all indicated comparisons. The data underlying this Figure can be found in https://doi.org/10.5281/zenodo.18250101.

## *CAP1* missense and nonsense mutations are recurrently found in clinical isolates

Identifying mutations that contribute to drug resistance in *C. albicans* clinical isolates remains challenging due to the substantial sequence variation between isolates [40–43]. To investigate the occurrence of *CAP1* mutations in clinical isolates, we performed variant calling on 300 *C. albicans* clinical isolate genomes, including 101 bloodstream isolates we obtained from five regional hospitals [5,44] and 199 publicly available genomes from two previous studies [40,41]. From the 300 clinical isolates, we identified 25 distinct *CAP1* missense and nonsense mutations relative to the SC5314 reference sequence, including 18 different missense mutations, four different insertions within the polyglutamine (polyQ) tract centered around amino acid position 184, and three nonsense mutations (Q49*, Q298*, and E448*) (Fig 2A and S2 Data). Six of the 25 mutations occurred within the b-Zip domain (Fig 2A).

Strikingly, the bloodstream isolate MEC079 contained a *CAP1* nonsense mutation E448* that was only two amino acids away from the nonsense mutation C446* we detected *in vitro* (Fig 2A). Both E448* and C446* were inside the first CSE repeat of the C-terminal cysteine-rich domain (C-CRD), and protein structure prediction indicated that these mutations resulted in a C-terminal truncation, disrupted the helix-loop structures of the C-CRD, and removed the nuclear export signal (Figs 2A, S2A Fig and S2B Fig). The other two heterozygous nonsense mutations (Q49* and Q298*), both of which occurred on the same allele in bloodstream isolates MEC218 and MEC219, were identified within the bZip domain or close to the N-CRD, and resulted in disruption of protein structure, likely resulting in Cap1 loss-of-function (Figs 2A and S2B Fig). Strains harboring these three nonsense mutations were isolated from two patients who, according to retrospective review, had not received azole antifungals prior to isolation of the *CAP1* mutant strains. However, isolates MEC218 and MEC219 from patient 1 were collected two months after an 18-day course of the echinocandin micafungin. Isolate MEC079 was obtained from a patient (patient 2) who had no known antifungal drug exposure prior to sample collection. Taken together, *CAP1* nonsense mutations are recurrently identified in isolates causing bloodstream infections even in the absence of azole treatment.

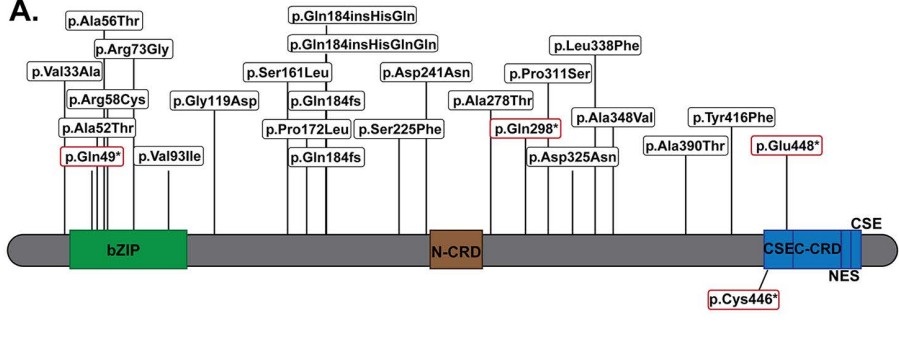

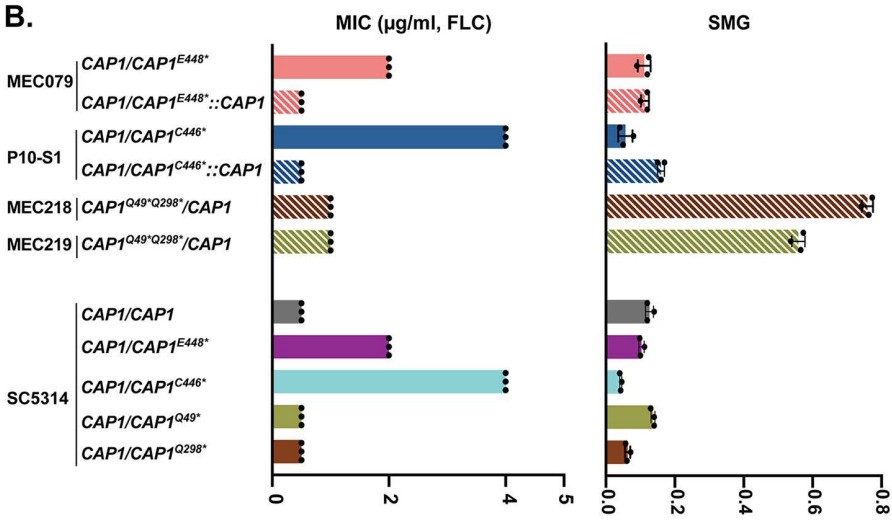

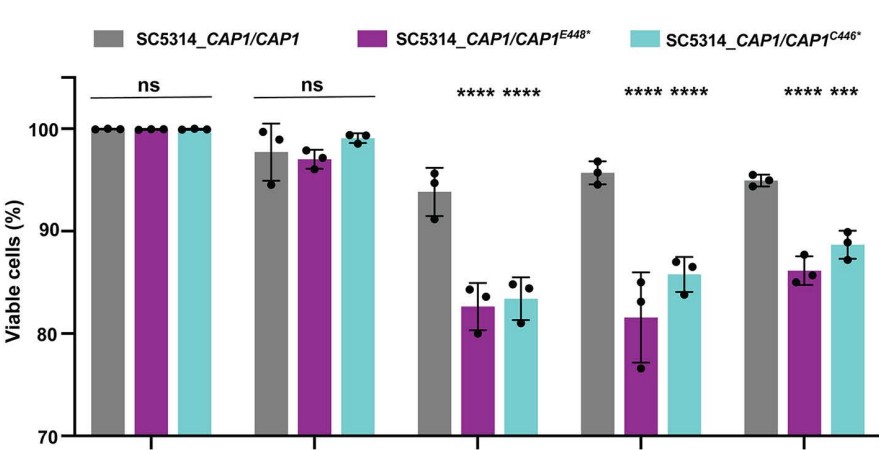

**Fig 2. C-terminally truncated Cap1 results in increased MIC but decreased cell viability. A.** *CAP1* sequence variants. The position of all missense and nonsense variants detected across 300 clinical isolates and an *in-vitro* evolution experiment, relative to the SC5314 reference genome. All insertions and deletions that occurred within the polyglutamine tract, centered around Gln184, are presented as frameshifts (Gln184fs). Nonsense mutations are indicated with an asterisk and a red box. Cap1 protein domains/features are indicated: the b-Zip domain, the Cysteine Rich Domains (CRD) at the N-terminus (N-CRD) and C-terminus (C-CRD), the nuclear export signal (NES), and Cysteine, Serine, Glutamic acid repeat (CSE). The data underlying this figure can be found in S2 Data. **B.** FLC susceptibility assay for MIC at 24 h (left, µg/ml) and tolerance or SMG at 48 h (right). The top half includes clinical isolate MEC079 (*CAP1/CAP1^E448*^*), the MEC079 strain with its mutated allele replaced with wild-type allele (*CAP1/CAP1^E448*^::CAP1*), the

FLC-evolved mutant P10-S1 (*CAP1*/*CAP1^C446\**), the P10-S1 strain with its mutated allele replaced with the wild-type allele (*CAP1*/*CAP1^C446\**::*CAP1*), and clinical isolates MEC218 and MEC219 (*CAP1^Q49\*Q298\**/*CAP1*). The bottom half includes all mutations engineered into the SC5314 genetic background (*CAP1*/*CAP1^C446\**, *CAP1*/*CAP1^E448\**, *CAP1*/*CAP1^Q49\**, *CAP1*/*CAP1^Q298\**, with the wild-type strain SC5314 (*CAP1*/*CAP1*) as the control. For MIC values, each dot represents a single replicate, and each bar represents the average of three biological replicates of a single strain; SMG values are mean ± SD calculated from three biological replicates of a single strain. The data underlying this Figure can be found in https://doi.org/10.5281/zenodo.18250101. **C.** Cell viability post-FLC MIC assay of two *CAP1* mutants engineered into the SC5314 background (*CAP1*/*CAP1^E448\** and *CAP1*/*CAP1^C446\**) compared to SC5314 from Fig 2B. Cell viability was determined by propidium iodide staining post-FLC MIC assay at 0, 1, 4, 8 and 16 µg/ml FLC (all raw data, as well as 0.5 and 2 µg/ml FLC, are provided in S3 Data). Data were assessed for normality with a Shapiro–Wilk test, and significant differences between the *CAP1* mutant and wild-type control across different FLC concentrations were calculated using a two-way ANOVA with Dunnett's multiple comparisons test (two-sided); *** $P < 0.001$, ****$P < 0.0001$, ns $P > 0.05$; the exact $P$ value is ***0.0005 for the indicated comparisons. Values are mean ± SD calculated from three biological replicates. The data underlying this figure can be found in S3 Data.

## C-terminal truncation of *CAP1* results in an increased fluconazole MIC and decreased cell viability at concentrations above the MIC

To determine the impact of *CAP1* nonsense mutations on FLC susceptibility, we measured both the minimum inhibitory concentration (MIC) and supra-MIC growth (SMG) of all isolates with *CAP1* nonsense mutations, including three clinical isolates and one *in vitro*-evolved isolate. Relative to wild-type, the *in vitro*-evolved strain P10-S1 (*CAP1*/*CAP1^C446\**) had the strongest increase in FLC MIC (8-fold increase), followed by clinical isolate MEC079 (*CAP1*/*CAP1^E448\**) (4-fold increase), and isolates MEC218 and MEC219 (*CAP1^Q49\*Q298\**/*CAP1*) (2-fold increase) (Fig 2B). Isolates MEC218 and MEC219 also had high FLC tolerance, measured by the supra-MIC growth (SMG of 0.66, Fig 2B).

Next, we engineered all four *CAP1* nonsense mutations into the drug-sensitive wild-type background (SC5314). The MIC of the engineered mutants *CAP1*/*CAP1^C446\** and *CAP1*/*CAP1^E448\** was identical to the MIC of the corresponding evolved mutant and clinical isolate (Fig 2B). We also replaced the mutated *CAP1* alleles of MEC079 (*CAP1*/*CAP1^E448\**::- *CAP1*) and P10-S1 (*CAP1*/*CAP1^C446\**::*CAP1*) with the wild-type *CAP1* allele, and this allele replacement decreased the MIC to wild-type SC5314 levels (Fig 2B). These results support that heterozygous *CAP1* C-terminal truncation caused by C446* or E448* causes decreased drug susceptibility.

In contrast, when the nonsense mutations located closer to the N-terminus were engineered into SC5314 (*CAP1*/*CAP1^Q49\** and *CAP1*/*CAP1^Q298\**), they had no effect on FLC susceptibility (Fig 2B). Additionally, we engineered *CAP1* overexpression, heterozygous deletion, and homozygous null deletion strains and found that they were viable and had no impact on FLC susceptibility in the SC5314 background (S2C Fig). Importantly, after multiple attempts, we were unable to generate homozygous *CAP1* C-terminal truncations, indicating that these genotypes are not viable. Taken together, we conclude that C-terminal truncation of *CAP1* reduces drug susceptibility in *C. albicans*, and this is not simply caused by loss-of-function or overexpression of *CAP1*.

Next, we measured cell viability after FLC treatment using both YPAD spot plates and quantification of propidium iodide-stained cells. Almost no viable cells were recovered from the *CAP1* C-terminal truncation mutants after treatment with >16 µg/ml FLC (S3A and S3B Fig). Compared to the wild-type control, engineered strains carrying a *CAP1* C-terminal truncation exhibited significantly decreased cell viability (about 15%) after treatment with FLC concentrations above their MIC, includ- ing 4, 8, 16 µg/ml FLC ($P < 0.01$, Fig 2C), but not at lower FLC concentrations, including 0.5, 1, 2 µg/ml FLC (Fig 2C and S3 Data). Allele replacement with wild-type *CAP1* rescued the viability defect of the *in vitro*-evolved *CAP1* mutant (P10- S1_*CAP1*/*CAP1^C446\**) in FLC (Figs 2C, S3A Fig and S3C Fig). However, no decrease in viability was observed for clini- cal isolate MEC079 relative to SC5314 or its mutated allele-replaced strain (S3B and S3D Fig), while the same mutation engineered into the SC5314 (SC5314_ *CAP1*/*CAP1^E448\**) was fungicidal at high concentrations of FLC (Figs 2C and S3B Fig). We attribute the survival of MEC079 at high concentrations of FLC to one or more of the ~11,000 SNP differences (allele frequency >0.3) between SC5314 and MEC079, including ~100 nonsynonymous variants within Cap1 gene targets that are related to oxidative stress response (e.g., *CIP1, GCS1,* and *GLR1*). These viability results suggests that high

concentrations of FLC are fungicidal to SC5314 cells carrying a *CAP1* C-terminal truncation, while clinical isolate MEC079 has acquired additional sequence variants that appear to rescue this killing effect.

**High levels of oxidative stress induce fungicidal effects of fluconazole in mutants with a *CAP1* C-terminal truncation**

Given the regulation of *CAP1* on OSR and azoles combined with stimulation of superoxide increases the fungicidal effect against *C. albicans* [15,45], we hypothesized that *CAP1* C-terminal truncation might contribute to the fungicidal action of FLC through stimulation of the oxidative stress response (OSR). To assess OSR, we quantified the growth rate of all *CAP1* C-terminal truncation mutants under increasing concentrations of $H_2O_2$ and 4-Nitroquinoline N-oxide (4NQO), a superoxide-generating agent. Compared to the wild-type control, all mutants with a *CAP1* C-terminal truncation had decreased susceptibility to high concentrations of 4NQO (0.25 and 0.3 μg/ml) but no change in $H_2O_2$ (S3E Fig).

While antifungal drugs can have both synergistic and antagonistic interactions with other drugs, the extent to which different genetic backgrounds modulate these interactions remains poorly understood. We used a checkerboard assay to determine the combined effects of increasing concentrations of FLC (0−256 μg/ml) and 4NQO (0–0.5 μg/ml) on the growth rate of two engineered *CAP1* mutants (*CAP1/CAP1^{C446\*}* and *CAP1/CAP1^{E448\*}*) and the wild-type control (*CAP1/CAP1*) (Figs 3A, 3B and S4A Fig). Only high concentrations of FLC (≥4 μg/ml) combined with high concentrations of 4NQO (0.25 μg/ml) exhibited a killing effect on the wild-type control (Fig 3A). In contrast, high concentrations of FLC were sufficient to cause killing of the *CAP1* mutants, even without the addition of 4NQO (Figs 3B and S4A Fig).

To determine if the FLC killing effect was related to intracellular ROS generation, we quantified intracellular ROS levels at different FLC concentrations using a fluorescent ROS probe. In the wild-type strain, the intracellular ROS levels increased dramatically from 0 to 0.5 μg/ml FLC, continued to increase from 0.5 to 1 μg/ml FLC, and then plateaued (Figs 3C, 3D, and S4B Fig). In contrast, the two engineered *CAP1* mutants maintained low intracellular ROS levels at 0–1 μg/ml FLC, followed by an exponential increase in ROS levels at 1 and 2 μg/ml FLC to levels that were approximately 2.0-fold higher than wild-type (Figs 3C, 3D and S4B Fig). Notably, the steepest increase in intracellular ROS levels for each strain coincided with its respective FLC MIC (0.5 μg/ml FLC for wild-type, 2 μg/ml FLC for the *CAP1* mutants) (Figs 3C, 3D, and S4B Fig). These findings reveal that the C-terminal truncation of *CAP1* fundamentally alters the dose-response relationship between FLC exposure and the generation of oxidative stress. Specifically, *CAP1* truncation confers protection against ROS accumulation at sub-inhibitory FLC concentrations, while paradoxically sensitizing cells to excessive ROS production at supra-MIC concentrations. The additional correlation between elevated ROS levels and decreased viability suggested that oxidative stress might be directly responsible for the fungicidal effect observed in *CAP1* mutants.

To directly determine whether the increased ROS accumulation in *CAP1* C-terminally truncated mutants was responsible for their decreased cell viability in the presence of FLC, we supplemented a FLC MIC assay with a ROS inhibitor (N-acetyl-L-cysteine). Addition of the ROS inhibitor rescued the viability of *CAP1* mutants at 4, 8, and 16 μg/ml FLC, while the FLC MIC did not change (Figs 3E, 3F, S4C Fig and S4D Fig and S3 Data). This demonstrates that FLC-induced oxidative stress has fungicidal activity against *CAP1* mutants with a heterozygous C-terminal truncation, establishing a mechanistic link between *CAP1* truncation, dysregulated ROS homeostasis, and enhanced fungicidal activity.

***CAP1* C-terminal truncation leads to nuclear retention and transcriptional activation**

To elucidate the molecular mechanisms underlying the altered OSR and altered drug susceptibility in *CAP1* C-terminally truncated mutants, we performed transcriptional analysis of wild-type SC5314 (*CAP1/CAP1*) and one engineered mutant (*CAP1/CAP1^{C446\*}*) using RNA-seq in two different log phase conditions: YPAD and YPAD+2 μg/ml FLC. Overall, *CAP1/CAP1^{C446\*}* had altered gene expression from the wild-type control in both YPAD and FLC (Fig 4A). We performed differential expression analysis to identify genes with a significant change in expression in *CAP1/CAP1^{C446\*}* compared to the

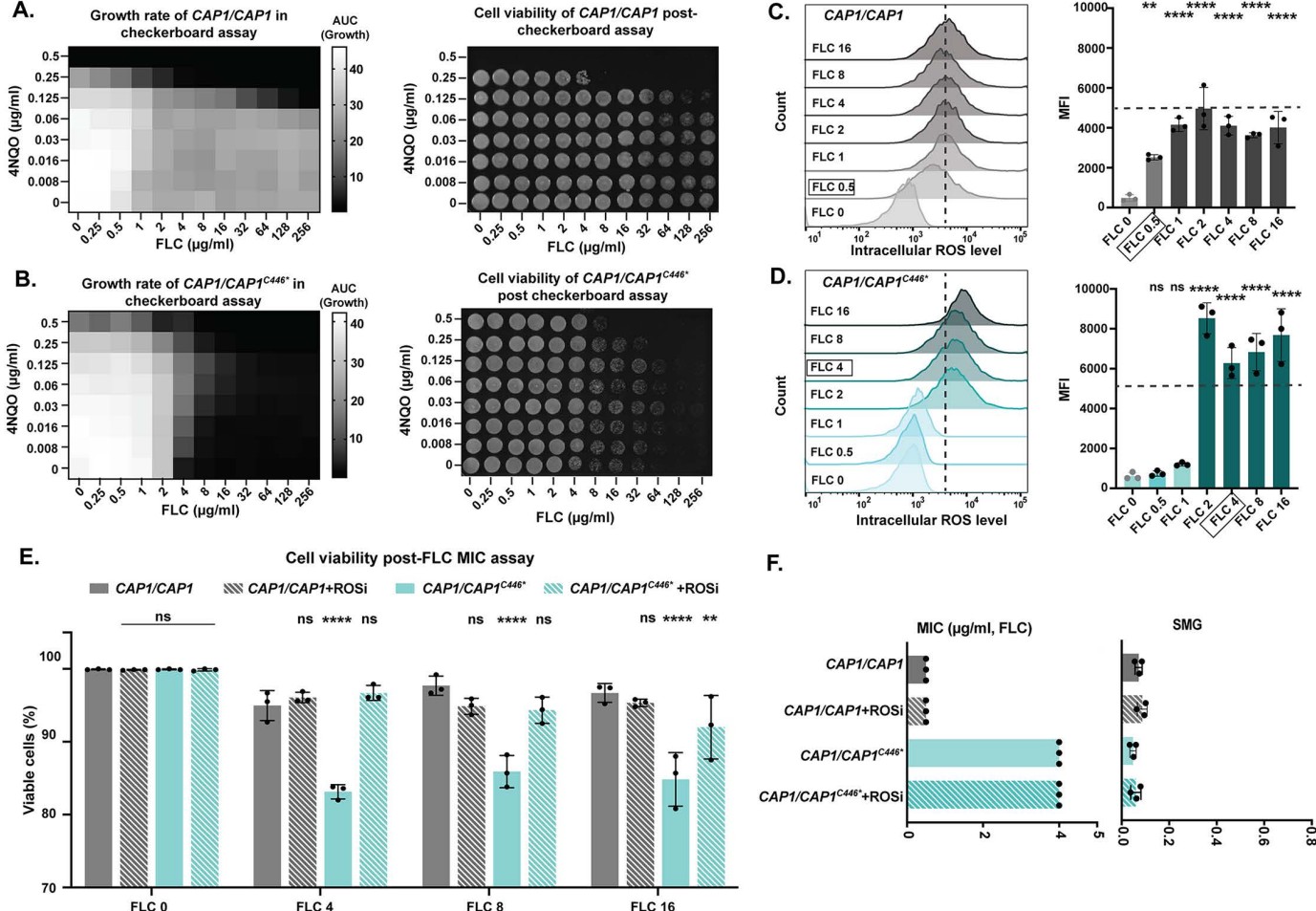

**Fig 3. High levels of intracellular ROS in the *CAP1* C-terminal truncation mutant cause decreased cell viability. A&B.** Checkerboard assay quantifying growth rate in the presence of increasing concentrations of FLC (X-axis, 0–256 µg/ml, 2-fold dilutions) and/or increasing concentrations of 4NQO (Y-axis, 0–0.5 µg/ml, 2-fold dilutions). Growth curves (left) and cell viability after the growth curve (right) of (**A**) the wild-type (*CAP1/CAP1*) and (**B**) *CAP1/CAP1^C446\** strains. Growth rate was quantified using the area under the curve (AUC heatmap) of the 48 h growth curve. Cell viability: cells from the growth curve were plated on YPAD agar and imaged after 24 h incubation. **C&D.** Intracellular ROS level of (**C**) wild-type (*CAP1/CAP1*) and (**D**) *CAP1/CAP1^C446\** strains at different concentrations of FLC (FLC0-FLC16, µg/ml). Intracellular ROS was determined with an ROS fluorescent detection kit combined with flow cytometry (methods). Left: Histogram of ROS fluorescence intensity from one representative biological replicate; Right: Median fluorescence intensity (MFI) of ROS and values are mean±SD were calculated from three biological replicates. Comparison was between FLC-exposed cells (FLC 0.5-FLC 16, µg/ml) and no drug control (FLC 0) for each strain. Dotted lines indicate the mean of highest-level intracellular ROS in the wild-type strain. Black boxes indicate the MIC of each strain. **E**. The proportion of viable cells for wild-type (*CAP1/CAP1*) and *CAP1/CAP1^C446\** strains post-FLC MIC assay at different concentrations of FLC (FLC 0-FLC 16, µg/ml) with or without the addition of an ROS inhibitor (S3 Data). Cell viability was determined by propidium iodide staining (methods). Statistical comparisons were against the FLC-only treated wild-type cells across different FLC concentrations. The data underlying this Figure can be found in S3 Data. **C&D&E**: Data were assessed for normality with a Shapiro–Wilk test, and significant differences were determined using (**C&D**) a one-way ANOVA with Dunnett's multiple comparisons test and (**E**) a two-way ANOVA with Tukey's multiple comparisons test (two-sided); ** $P < 0.01$, **** $P < 0.0001$, ns $P > 0.05$; the exact $P$ values are **0.0025 and 0.0089 and **** < 0.0001. **F.** 24 h MIC (left, µg/ml) and 48 h SMG (right) in FLC with or without treatment with the ROS inhibitor for *CAP1/CAP1^C446\** with wild-type strain (*CAP1/CAP1*) as the control. For MIC values, each dot represents a single replicate, and each bar represents the average of three biological replicates of a single strain; SMG values are mean±SD calculated from three biological replicates of a single strain. The data underlying this figure (A-D&F) can be found in https://doi.org/10.5281/zenodo.18250101.

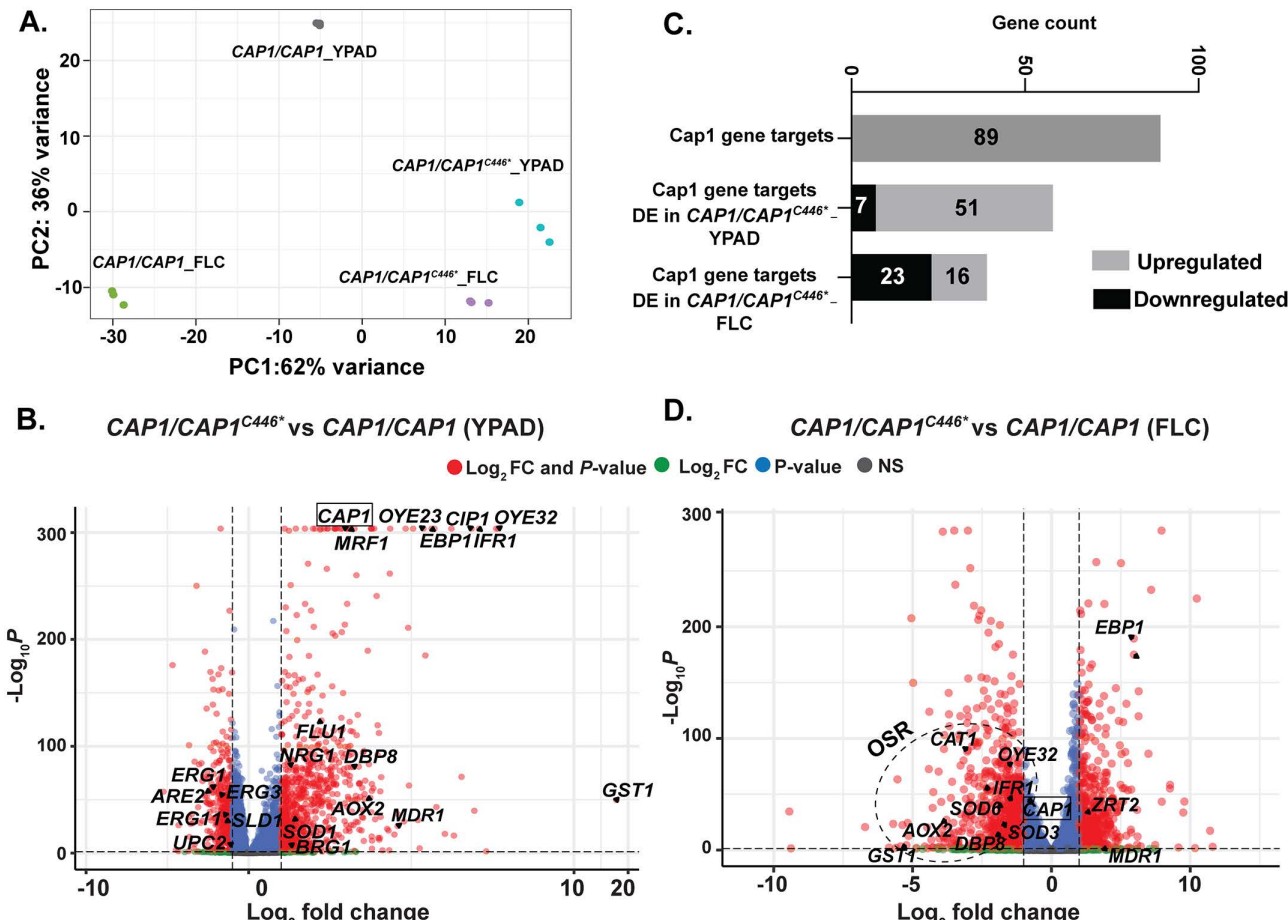

**Fig 4.** *CAP1* **C-terminally truncated mutants exhibited upregulated drug transport and downregulated OSR with fluconazole exposure. A.** Principal component analysis of transcriptional data in YPAD and YPAD+ 2 µg/ml FLC for *CAP1/CAP1* and *CAP1/CAP1^C446\**. **B**. Volcano plot for differentially expressed (DE) genes (log2 fold change ≥ 1 or ≤ −1 and adjusted p-value < 0.05) in the *CAP1/CAP1^C446\** mutant compared to *CAP1/CAP1* in YPAD. The data underlying this Figure can be found in S4 Data. **C**. The number of Cap1 gene targets (total 89) that were upregulated or downregulated in *CAP1/CAP1^C446\** relative to *CAP1/CAP1* in YPAD and FLC. The data underlying this Figure can be found in https://doi.org/10.5281/zenodo.18250101. **D**. Volcano plot for differentially expressed genes (log2 fold change ≥ 1 or ≤ −1 and adjusted p-value < 0.05) in the *CAP1/CAP1^C446\** mutant compared to *CAP1/CAP1* in 2 µg/ml FLC. **B&D**: Genes that are significantly differentially expressed by both fold change and p-value cut-offs are in red. The data underlying this Figure can be found in S7 Data.

wild-type strain in both conditions (absolute $\log_2$ fold change ≥ 1 and adjusted *P*-value < 0.05). In YPAD, *CAP1* C-terminal truncation resulted in widespread transcriptional changes, with 461 genes significantly downregulated and 762 genes significantly upregulated (including *CAP1* itself) compared to wild-type (Fig 4B and S4 Data). Within the downregulated gene set, we found an overrepresentation of genes encoding ergosterol and sphingolipid biosynthesis/transport proteins (e.g., *UPC2, ARE2, ERG1, ERG3, ERG11, OSH2,* and *SLD1*) relative to all expressed genes (Fig 4B and S5 Data). In addition to azole resistance, the *CAP1/CAP1^C446\** mutant had reduced filamentation compared to wild-type, and this corresponded with upregulation of genes encoding two transcriptional repressors of filamentation, *BRG1* and *NRG1*, as well as their downstream targets, among the 762 upregulated genes (Figs 4B and S5A Fig-S5C Fig and S4 Data).

Among all differentially expressed genes, 58 of the 89 known Cap1 gene targets [15] had altered expression in the *CAP1/CAP1^C446\** mutant compared to wild-type, with 51 upregulated and 7 downregulated (Fig 4C). Gene Ontology

analyses of the 51 upregulated Cap1 targets revealed an overrepresentation of genes associated with the OSR, including *CAP1* itself, *GLR1*, *CIP1*, *AOX2, OYE32,* and *SOD1*, and drug transport, including *MDR1* and *FLU1* ([Figs 4B](), [4C]() and [S6A Fig]() and [S6 Data]()). This indicates that C-terminal truncation of Cap1 results in the transcriptional activation of *CAP1* and upregulation of genes involved in the OSR and drug transport, even in the absence of drug.

   *CAP1* C-terminal truncation (C446*) had markedly different effects on gene expression in the presence of FLC as compared to wild-type. *CAP1* C-terminal truncation in combination with FLC exposure led to downregulation of 785 genes and upregulation of 533 genes compared to wild-type in FLC ([Figs 4D]() and [S4B Fig]() and [S7 Data]()). Notably, several genes that were highly upregulated in YPAD were downregulated with FLC exposure, including the highest upregulated gene *GST1* ([Figs 4D]() and [S4B Fig]() and [S7 Data]()). Among the 89 known Cap1 gene targets, 39 were differentially expressed in the presence of FLC, with 23 downregulated and 16 upregulated ([Fig 4C]() and [4D]()). Importantly, many of the downregulated genes were associated with the oxidative stress response, including *CAP1, CAT1, CIP1, SOD6*, *OYE32*, and *AOX2* ([S6B Fig]() and [S8 Data]()). In contrast, drug transporters like *MDR1* were still upregulated in the *CAP1* mutant in the presence of FLC ([Fig 4D]()).

   To dissect the mechanism underlying the downregulation of the OSR in *CAP1* mutant cells, we quantified the relative fold induction of Cap1 gene targets in the mutant or wild-type cells in response to FLC. The *CAP1/CAP1^{C446*}* mutant exhibited a substantially more neutral transcriptional response of OSR genes compared to wild-type cells, in response to FLC ([S6C Fig]()). This finding indicates that the downregulation of OSR genes in the *CAP1* mutant was primarily due to its impaired ability to mount an OSR in response to FLC, in contrast to the robust induction of the OSR observed in wild-type cells. Additionally, *MDR1,* which encodes a drug transporter, was induced further in the *CAP1/CAP1^{C446*}* mutant in response to FLC, in contrast to no *MDR1* induction in the wild-type cells in response to FLC ([S6C Fig]()), suggesting that induction of this drug efflux pump was specific to the *CAP1*-truncation mutant.

   Using fluorescent tagging, we independently followed both the truncated and wild-type proteins in both the *CAP1* mutant and wild-type cells. In YPAD, the truncated Cap1 proteins (C446* and E448*) had co-localization with Hoechst nuclear staining, whereas the wild-type protein in wild-type cells had diffuse cytoplasmic localization ([Figs 5A]() and [S6D Fig](), GFP). The mutated Cap1 protein also influenced the subcellular localization of wild-type Cap1. In YPAD, the fluorescently tagged wild-type Cap1 protein exhibited punctate localization adjacent to the nucleus in the *CAP1^{C446*}/CAP1-RFP* cells ([Fig 5A](), RFP), but diffuse cytoplasmic distribution in the *CAP1-GFP/CAP1* wild-type cells ([Fig 5A](), GFP top panel). FLC exposure induced abnormal nuclear and cell morphologies, which interfered somewhat with the nuclear localization of the truncated Cap1 protein, resulting in its accumulation adjacent to punctate nuclei ([Fig 5B]()). In contrast, the wild-type Cap1 protein in wild-type cells mostly maintained its diffuse cytoplasmic localization with a few puncta that seem to colocalize with nuclei ([Fig 5B]()).

   In addition to Cap1 localization, we determined the effect of the C-terminal truncation on protein stability. We used flow cytometry to quantify the fluorescent intensity of wild-type and mutant Cap1-GFP protein under a constitutive promoter during exponential growth in YPAD. Cap1-GFP with a C-terminal truncation (C446* and E448*) exhibited significantly decreased fluorescent intensity compared to the wild-type Cap1-GFP ([Fig 5C](), $P<0.0001$). These results indicate that the C-terminal truncation could directly impact Cap1 protein stability as observed previously [15,17]. Notably, the E448* truncation had significantly reduced fluorescent intensity relative to the C446* truncation ([Fig 5C](), $P<0.001$), further indicating that the E448* truncation resulted in less Cap1 protein stability than the C446* truncation.

   In summary, the transcriptional, subcellular localization, and protein stability analyses revealed a dual regulatory mechanism governing Cap1 function in the absence or presence of antifungal drug. In the absence of FLC, a Cap1 C-terminal truncation constitutively activates the Cap1 transcriptional network through nuclear retention, resulting in upregulation of both oxidative and drug stress responses, despite the reduced protein stability ([Fig 6A]()). In the presence of FLC, the Cap1-dependent transcriptional regulatory network was disrupted, resulting in an upregulation of drug transporter expression and loss of an oxidative stress response. This azole-induced inhibition of Cap1 activity parallels previous findings

                                                                      

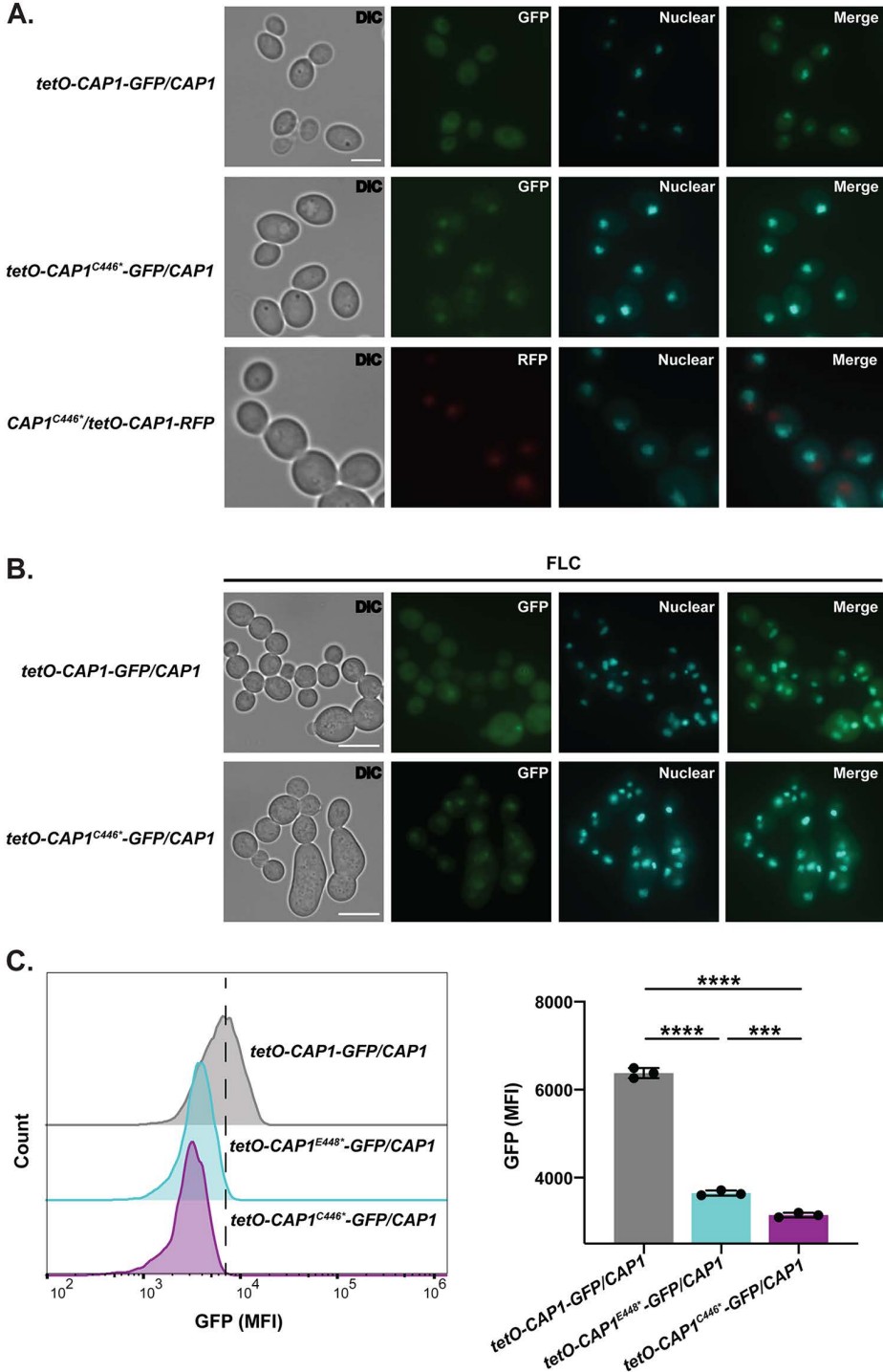

**Fig 5. Subcellular localization and fluorescent intensity of wild-type Cap1 and Cap1-GFP with C446*. A.** Subcellular localization of Cap1-GFP in wild-type background (top, *tetO-CAP1-GFP/CAP1*), Cap1-GFP with C446* (middle, *TeO-CAP1^{C446*}-GFP/CAP1*) and wild-type Cap1-RFP (bottom, *CAP1^{C446*}/TeO-CAP1-RFP*) in the *CAP1/CAP1^{C446*}* background. Scale bar, 5 μm. **B.** Subcellular localization of wild-type Cap1-GFP (top, *TetO-CAP1-GFP/CAP1*) and Cap1 with C446* (bottom, *TetO-CAP1^{C446*}-GFP/CAP1*) in the SC5314 background with 2 μg/ml FLC exposure. Scale bar, 10 μm. **A&B**: Hoechst (light blue) was applied, indicating nuclear co-localization (Methods). **C**. GFP fluorescent intensity of Cap1-GFP, Cap1-GFP with C466*, and Cap1-GFP with E448* in the wild-type background measured using flow cytometry (methods). Left: Histogram of GFP fluorescence intensity from one

representative biological replicate, where the dotted line indicates the mean fluorescence intensity in the wild-type strain. Right: Median fluorescence intensity (MFI) of GFP where the mean ± SD were calculated from three biological replicates. Data were assessed for normality with a Shapiro–Wilk test, and significant differences were determined using a one-way ANOVA with Tukey's multiple comparisons test and ****$P<0.0001$, ***$P<0.001$; the exact $P$ value is ***0.0007. Comparison was between each strain. The data underlying this Figure can be found in https://doi.org/10.5281/zenodo.18250101.

where combined oxidative and osmotic stress led to intracellular ROS accumulation, which similarly suppressed Cap1 function, eliminated oxidative stress–responsive transcription, and ultimately caused fungal cell death [22,46].

### C-terminally truncated *CAP1* mediates fluconazole-specific drug response

We tested whether the *CAP1* C-terminal truncation mutants also exhibit increased resistance to other common antifungal drugs and if other azoles can pose similar fungicidal effects on these mutants. The *CAP1* C-terminal truncation mutants (*CAP1*/*CAP1^C446*^* and *CAP1*/*CAP1^E448*^*) had the same MIC as the wild-type control for antifungal drugs micafungin, voriconazole, itraconazole, and miconazole (S7A Fig). Notably, the *CAP1* C-terminal truncation mutants had an increased MIC to cerulenin, an inhibitor of both fatty acid synthase and sterol synthesis, and substrate of the Mdr1p efflux pump (8-fold and 4-fold increase for *CAP1*/ *CAP1^C446*^* and *CAP1*/ *CAP1^E448*^*) (Figs 6B and S7C Fig). However, the *CAP1* C-terminal truncation mutants had decreased viability at supra-MIC concentrations of all three azoles, similar to FLC (Figs 6C and S7B Fig). This suggests that *CAP1* C-terminal truncation increases resistance to cerulenin and increases the fungicidal activities of azoles more broadly.

Next, we tested if the decreased drug susceptibility of the *CAP1* mutants was caused by overexpression of *MDR1*. Deletion of *MDR1* from the *in vitro*-evolved *CAP1* mutant (P10-S1_*mdr1*Δ/Δ) decreased the FLC MIC from 4 µg/ml to 1 µg/ml, while deletion of *MDR1* from the wild-type SC5314 had no impact on FLC susceptibility (Figs 6D and S7D Fig). Overexpression of *MDR1* increased the FLC MIC from 0.5 µg/ml to 2 µg/ml [39] and the cerulenin MIC from 0.5 µg/ml to 6 µg/ml , but had no impact on drug susceptibility in other azoles or micafungin in the SC5314 background (Figs 5B and S7A Fig). We conclude that upregulation of *MDR1* is the major driver for the decreased drug susceptibility of *CAP1* mutants in FLC and cerulenin.

### Fluconazole-selected *CAP1* C-terminal truncation leads to drug resistance in a murine model

We next determined the effect of the *CAP1* C-terminal truncation during a murine model of systemic infection with or without FLC treatment. Our identification of *CAP1* mutations that caused altered oxidative stress response in clinical isolates raised the question of whether these mutations contribute to fungal pathogenicity. Using a tail-vein injection model of disseminated candidiasis, we compared two engineered *CAP1* mutants (*CAP1*/*CAP1^C446*^* and *CAP1*/*CAP1^E448*^*) to a wild-type control carrying the same selective marker as *CAP1* mutants (*CAP1*/*CAP1*). By 15 days post-infection, all infected mice had died with no survival difference between the wild-type control and two *CAP1* C-terminally truncated mutants (Fig 7), demonstrating *CAP1* C-terminal truncation does not impair pathogenicity in a murine model.

Additionally, to determine if *CAP1* mutants also exhibit FLC resistance *in vivo*, three separate groups of infected mice were treated with 2 mg/kg of FLC from day 1 to day 5 post-infection, and survival was monitored until day 21. The 2 mg/kg FLC is a moderate dose resulting in decreased fungal burden without clearance of fungal cells [39]. With 2 mg/kg FLC treatment, mice infected with *CAP1*/*CAP1^C446*^* were dead before day 15 and had significantly shorter survival compared to the wild-type control and *CAP1*/*CAP1^E448*^* ($P=0.0064$ and 0.0017, Log-rank (Mantel-Cox) test) (Fig 7A). This indicates that the nonsense mutation at C446 confers drug resistance *in vivo*. The C-terminal truncation at C446 had a greater impact than E448 on FLC resistance *in vivo*, consistent with a 2-fold higher FLC MIC *in vitro* (Fig 3B), and indicates that truncation at the cystine (C446 of the CSE protein domain) is driving these phenotypic differences. To determine the mechanism underlying the observed phenotypic differences, we applied RT-qPCR to compare the transcriptional level of major drug transporter *MDR1* and *CAP1* in cells with truncation at C446 or E448 cultured *in vitro*. Both mutants had elevated

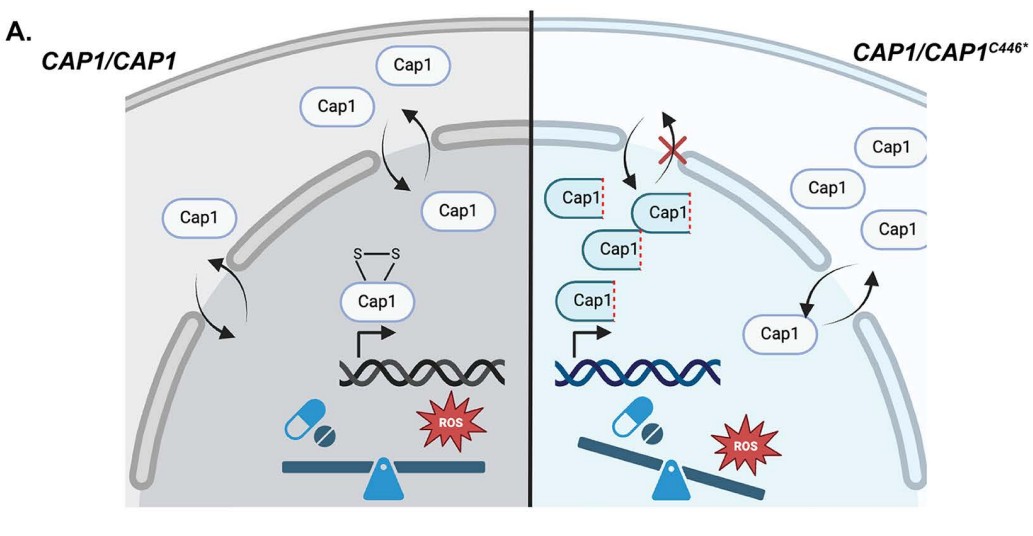

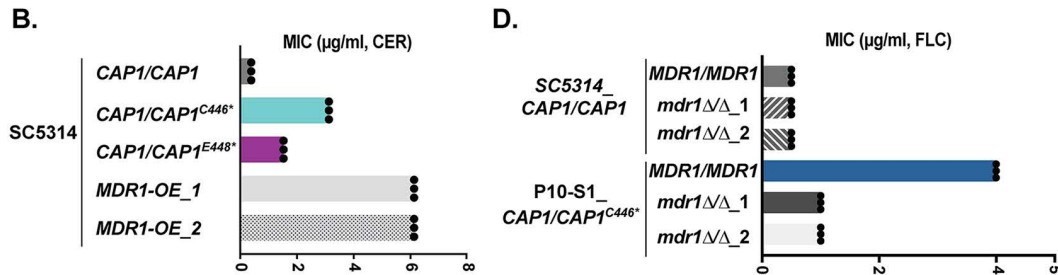

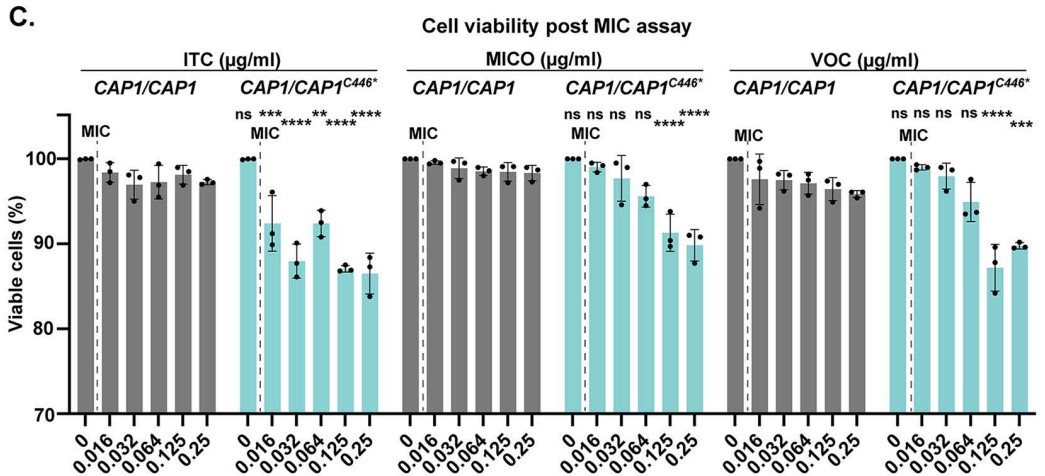

**Fig 6. C-terminally truncated *CAP1* leads to increased MIC only in fluconazole but induces fungicidal effects of multiple azoles. A.** Schematic of subcellular localization and transcription activation of *CAP1* in wild-type (*CAP1/CAP1*, left) and *CAP1/CAP1^C446*^*. Figure generated with BioRender. **B.** 24 h MIC (μg/ml) in cerulenin (CER) for strains *CAP1/CAP1^C446*^*, *CAP1/CAP1^E448*^* and *MDR1* overexpression strains (two independent transformants *MDR1-OE_1* and *MDR1-OE_2)*, with wild-type background strain (SC5314, *CAP1/CAP1*) as the control. **C.** The proportion of viable cells for wild-type (*CAP1/CAP1*) and *CAP1/CAP1^C446*^* post 48 h MIC assay at different concentrations of itraconazole (ITC), miconazole (MICO), and voriconazole (VOC) (0-0.25 μg/ml). Cell viability was determined by propidium iodide staining (methods). Comparison was between *CAP1/CAP1* and *CAP1/CAP1^C446*^* across different concentrations of drugs. Data were assessed for normality with a Shapiro–Wilk test, and significant differences using two-way ANOVA with Šídák's multiple comparisons test (two-sided); ** $P < 0.01$, *** $P < 0.001$, **** $P < 0.0001$, ns $P > 0.05$; the exact $P$ values are **0.0082, ***0.0010 and 0.0004, and ****<0.0001. **D.** 24 h MIC (μg/ml) in FLC for the wild-type, FLC-evolved mutant P10-S1 (*CAP1/CAP1^C446*^*), and two independent *MDR1* null mutants (*mdr1Δ/Δ_1&_2*) engineered into both the P10-S1 and the wild-type genetic backgrounds (*CAP1/CAP1*). **B&D:** For MIC values, each dot represents a single replicate, and each bar represents the average of three biological replicates of a single strain. The data underlying this Figure can be found in https://doi.org/10.5281/zenodo.18250101.

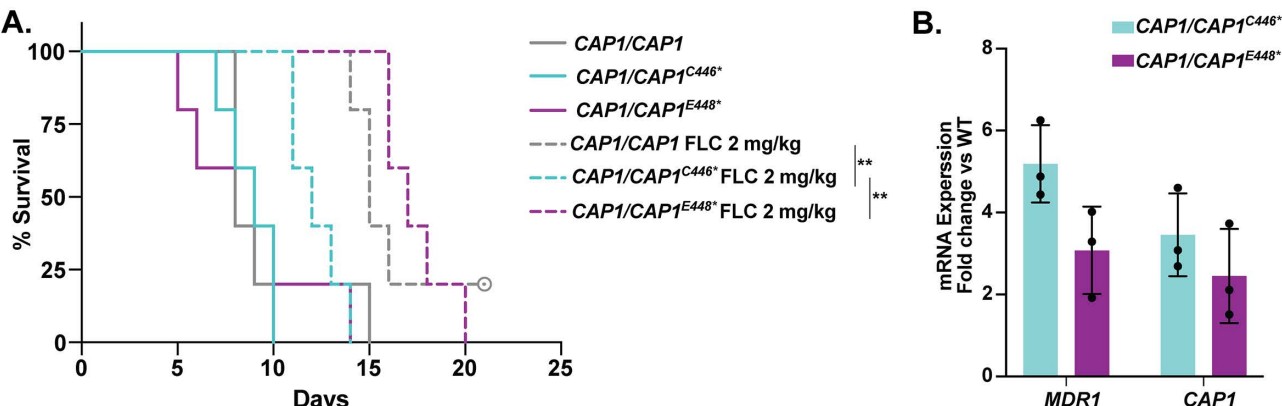

**Fig 7. Mutants with *CAP1* C-terminal truncation maintain their pathogenicity and the fluconazole-selected mutant exhibits drug resistance in a murine model. A.** Survival over time of ICR mice injected via the tail vein with 1 x 10⁵ cells of wild-type control (SC5314, *CAP1/CAP1-NAT*), *CAP1/CAP1^C446\**, and *CAP1/CAP1^E448\**. Five mice were used for each genotype and treatment. Mice were treated with PBS or 2 mg/kg FLC every day for 5 days post-infection and monitored daily for survival for 21 days. Significance differences between *CAP1/CAP1^C446\** and *CAP1/CAP1-NAT* or *CAP1/CAP1^E448\** using Log-rank (Mantel-Cox) test; **, *p*=0.0064 and 0.0017 respectively. **B.** mRNA expression quantified by RT-qPCR. The relative fold change (y-axis) of *MDR1* and *CAP1* in the engineered mutants *CAP1/CAP1^C446\** and *CAP1/CAP1^E448*, all relative to the wild-type SC5314 control, grown in YPAD media. Values are mean±SD calculated from three biological replicates. The data underlying this Figure can be found in https://doi.org/10.5281/zenodo.18250101.

expression of both *MDR1* and *CAP1* relative to wild-type. The mutant with the C446 truncation had ~1.5-fold higher *MDR1* expression than cells with the E448 truncation (5.2-fold and 3.5-fold, respectively Fig 7B), which likely contributes to the difference observed between the two truncation mutations *in vivo*.

## Discussion

Fungal pathogens utilize different stress response mechanisms during adaptation to their changing environments [9,47,48]. We provide a comprehensive analysis of *de novo* mutations identified in the conserved C-terminal CRD of the transcriptional regulator *CAP1* from *in vitro* and clinical isolates of *C. albicans*. Our work reveals how C-terminal truncation of *CAP1* causes azole resistance *in vitro* and in a mouse model of systemic infection, while also uncovering a previously unknown lethal interaction between *CAP1* truncation and supra-MIC concentrations of azole drugs, due to the formation of fungicidal ROS.

The Cap1 C-terminal truncation not only deletes the NES but also interrupts the interaction between Cap1 and its redox sensor, which directly impacts Cap1 transcriptional regulation. Cap1-mediated regulation of oxidative stress is governed by several regulatory proteins, including glutathione peroxidase (Gpx)-like-enzyme Gpx3, which serves as both sensor and transducer of the oxidative signal to *C. albicans* Cap1 and *S. cerevisiae* Yap1 [24,25,32]. During the oxidant process of Yap1, an intramolecular disulfide bond forms between Cys36 of Gpx3 and Cys598 of Yap1, resulting in the activated form of the transcriptional regulator [24,32]. Because the Cys598 of Yap1 aligns with the Cys446 in Cap1, we propose that disruption of Cys446 with a nonsense mutation impairs disulfide bond formation and thus transcriptional activation of Cap1 itself and its downstream targets (S2A Fig). Additionally, we found that *GPX3* RNA abundance was reduced in the *CAP1* C446 nonsense mutant (S5B Fig), which would further impair both the sensing of the oxidative signal and the transduction of the signal to Cap1.

The critical role of C446 in disulfide bond formation might also explain the small phenotypic differences observed *in vivo* and *in vitro* between the Cap1 mutants with C446* or E448*. Truncation at E448 might still permit disulfide bond formation, however, at substantially reduced efficiency, which paradoxically interferes with Cap1 activation. Indeed, the

E448* truncation mutant had reduced levels of *MDR1* transcription and a lower FLC MIC than the C446* mutation engineered in the same genetic background. Additionally, the E448* protein was less stable than the C446* protein, suggesting that truncation at E448 causes more pronounced negative effects on overall Cap1 function, potentially offsetting any gains in drug resistance. More work is needed to determine the effect of antifungal treatment on disulfide bond formation and the redox sensor process in both the wild-type and Cap1 truncation mutants.

A striking and unexpected finding of our study is that azoles at supra-MIC concentrations exert fungicidal effects on cells with a *CAP1* C-terminal truncation, contradicting previous reports that *CAP1* truncations exclusively enhance drug resistance [12,15,17,33]. This fungicidal conversion was directly linked to the production of azole-induced ROS and down-regulation of the oxidative stress response, revealing a previously unrecognized vulnerability in the resistant mutants. These findings help explain conflicting reports regarding FLC-induced ROS accumulation by demonstrating that mutational status fundamentally alters oxidative stress responses [34–37]. Future studies will investigate the cellular origins of azole-induced ROS and evaluate whether this oxidative vulnerability extends to other resistance mutations and fungal pathogens.

We report that *CAP1* C-terminal truncation had a broad impact on the transcription regulatory networks, impacting drug resistance and filamentation. Key transcription factors, including *MRR1*, *UPC2, ADR1*, *BRG1*, and *NRG1* all had increased expression upon *CAP1* C-terminal truncation (Figs 4B and S5A Fig). Drug resistance of *CAP1* mutants is driven by a complex mechanism. Previously, C-terminally truncated Cap1 (C333Δ) was found to cooperate with Mrr1 to regulate the induction of *MDR1,* resulting in increased drug efflux [17,20,21]. Similarly, we found induction of *MDR1* as well as other drug transporters, including *ZRT2* [49]. We also found activation of *UPC2 and ADR1,* which directly regulate ergosterol biosynthesis and fluconazole resistance [50,51], further supporting that Cap1 promotes azole resistance via multiple mechanisms. Furthermore, we found that Cap1 C-terminal truncation activated the transcriptional repressors of filamentation, Brg1 and Nrg1 [52]*,* which was associated with the filamentation defects of the Cap1 C-terminal truncation mutants (S5C Fig)*.* More work is needed to dissect the structure and complexity of regulatory networks mediating stress responses, including drug resistance, filamentation, and pathogenicity among fungal pathogens.

We propose that heterozygous nonsense *CAP1* variants exhibit distinct gene expression profiles during adaptation to the host immune response from either their homozygous or wild-type versions. Our analysis of 300 clinical isolates identified diverse heterozygous *CAP1* nonsense mutations, including both loss-of-function (Q49* and Q298*) and hyperactivation variants (E448*), all from patients without drug treatment history within the period of isolate collection. Using a mouse model of systemic infection, we demonstrated that heterozygous hyperactivated Cap1 enhanced pathogenicity during antifungal treatment, whereas the same mutant had no advantage without drug treatment. Previously, complete dysfunction (null deletion) of *CAP1* similarly had no impact on virulence using the same infection model used in our study, however, Cap1 was essential for macrophage survival, indicating that Cap1 is selectively important within different host environments [24]. Additionally, the recovery of heterozygous loss-of-function mutations from clinical isolates suggests that preservation of a single wild-type copy of Cap1 is important during bloodstream infection. One limitation of our and previous studies is that we did not determine the impact of the heterozygous null mutants (e.g., *CAP1*/*CAP1*$^{Q49*}$ or *CAP1*/*cap1*Δ) during interaction with macrophages or within a systemic murine model. Collectively, different *CAP1* mutations can be selected during bloodstream infection and hyperactivation mutations are beneficial during antifungal exposure. Future work is needed to dissect the interaction between heterozygous hyperactive and heterozygous dysfunction *CAP1* mutants and host immune stresses at different infection stages.

The selection of only heterozygous *CAP1* hyperactivation variants, from both *in vitro* and clinical isolates, suggests the detrimental effect of their homozygous status. *De novo* mutations in genes like *ERG11*, *MRR1,* and *EFG1* that cause drug resistance or reduced virulence are frequently homozygous [41,53–56]. Recently, we reported that heterozygous point mutations of *ERG251* cause azole tolerance in *C. albicans*. The heterozygous state of the *ERG251* point mutation persisted due to significant fitness defects caused by the homozygous mutants [49]. In the current study,

after multiple transformation attempts, we were not able to obtain homozygous C-terminal truncations of hyperactive *CAP1*, whereas homozygous null deletions of *CAP1* were viable. We found that the Cap1 C-terminal truncation alleles are retained in the nucleus in a heterozygous strain, even without exogenous oxidative stress activation. The inviable homozygous C-terminal truncations of Cap1 support that retention in the nucleus is detrimental to the cell, and maintaining heterozygosity is critical once the C-terminal truncation arises.

In summary, we provide new insights into the complex regulatory networks of Cap1 encompassing oxidative stress response and drug resistance in *C. albicans*, with important implications for understanding the emergence of antifungal resistance in both clinical and laboratory settings. The dissection of mechanisms underlying these regulatory networks directly contributes to our understanding of how this opportunistic pathogen adapts to host environments and therapeutic interventions throughout different environments.

## Methods

### Ethics statement

The mouse experiments were approved by the Institutional Animal Care and Use Committee of the Lundquist Institute for Biomedical Innovation at Harbor-University of California, Los Angeles Medical Center, under animal welfare assurance number D16-00213.

### Yeast isolates and culture conditions

All strains used in this study are listed in S1 Data, including *in vitro* evolved isolates, MEC clinical isolates, engineered yeast, and bacterial strains. Strains were stored at −80°C in 20% glycerol. Unless otherwise specified, isolates were grown in YPAD media (20g/L bactopeptone, 10g/L yeast extract, 2% dextrose, and 15 g/L agar for plates) supplemented with adenine (40 μg/ml) and uridine (80 μg/ml).

### Strain construction

Engineered strains were constructed by lithium acetate transformation using PCR products with at least 140 bp of homology to the target locus. All primers used in this study can be found in S1 Data.

(i) **Construct *CAP1* plasmid.** To construct *CAP1* allele replacement and point mutants, a plasmid carrying *CAP1-NAT* was created. The *CAP1* gene with upstream homology (1674 + 1675), *NAT* (1574 + 1575), and downstream *CAP1* (1676 + 1677) were fused into the pUC19 backbone (1578 + 1579). These PCR amplified fragments were ligated using NEBuilder HiFi assembly kit following manufacturer's instructions and transformed into the provided *E. coli*. Ampicillin-resistant transformants were screened by PCR for correct assembly using primer pairs 1352 + 1353 and saved in frozen stocks as pAS3148.

(ii) ***CAP1* point mutations.** Point mutations in *CAP1* were introduced by double-primer PCR. To create point mutation constructs, PCR primers carrying the mutation were paired with *CAP1* upstream and downstream primers to amplify a mutated *CAP1* construct from pAS3148. Four different point mutations were engineered, $CAP1^{Q298*}$ (1670 + 2071, 2070 + 1671), $CAP1^{Q49*}$ (1670 + 2023, 2022 + 1671), $CAP1^{C446*}$ (1670 + 1673, 1672 + 1671), and $CAP1^{E448*}$ (1670 + 2021, 2020 + 1671). The fragments were fused using SOE PCR to create full-length constructs that were transformed into the SC5314 background. NAT-resistant transformants were PCR screened for correct integration of the point mutation constructs at the *CAP1* locus using primers 1670 + 1575 (left of integration) and 1636 + 1671 (right of integration). Transformants were validated for correct integration and base substitution by whole-genome sequencing.

(iii) ***CAP1* mutant allele replacement.** Wild-type *CAP1-NAT* construct was PCR amplified from the *CAP1-NAT* plasmid (pAS3148) using primer pair 1670 + 1671 and transformed into background strains P10-S1_*CAP1/CAP1^{C446*}* and MEC079. NAT-resistant transformants were PCR screened for correct integration at the *CAP1* locus using primers

1670 + 1154 (left of integration) and 1636 + 1671 (right of integration). Transformants were validated for correct integration by whole-genome sequencing.

**(iv) *CAP1* overexpression.** The *TAR-TetO-NAT* promoter replacement was PCR amplified from pLC605, a generous gift from Leah E. Cowen [57], using primer pair 1666 + 1746 and transformed into the SC5314 background. NAT-resistant transformants were PCR screened for correct integration in front of the *CAP1* locus using primers 1747 + 1176 (left of integration) and 1177 + 1748 (right of integration).

**(vi) *CAP1* heterozygous deletion.** The *FLIP-NAT* construct with *CAP1* flanking sequence was PCR amplified from plasmid pJK863 [58] using primer pair 1666 + 1667 and transformed into the SC5314 background. NAT-resistant transformants were PCR screened for correct integration at the *CAP1* locus using primers 1670 + 1045 (left of integration) and 1636 + 1671 (right of integration). Transformants were validated for correct integration by whole-genome sequencing.

**(v) *CAP1* homozygous deletion.** Validated *CAP1/cap1Δ* transformants were inoculated in YNB + BSA from frozen stocks and cultured for 48 h at 220 rpm, 30°C to promote FLIP-mediated excision of *NAT.* Cultures were diluted and ~100 cells were plated on YPAD agar, then incubated for 24 h, 30°C. Colonies were patched to both YPAD and YPAD+150 µg/ml NAT agar. For colonies that were only able to grow on YPAD, PCR screening was performed to validate the correct excision of *FLIP-NAT* using primer pairs 1574 + 1575 (*NAT*) and 1670 + 1671 (across *CAP1*). Correct colonies were re-transformed with the *FLIP-NAT* with *CAP1* flanking sequence construct was amplified with primers 1666 + 1667 from plasmid pJK863 [58]. NAT-resistant transformants were PCR screened for correct integration at the *CAP1* locus using primers 1670 + 1045 (left of integration), 1636 + 1671 (right of integration), and 1678 + 1673 (inside *CAP1*). Transformants were validated for correct integration by whole-genome sequencing.

**(vi) *MDR1* heterozygous deletion.** The *FLIP-NAT* construct with *MDR1* flanking seqeunce was PCR amplified from plasmid pJK863 [58] using primers 1720 + 2000 and transformed into the P10-S1_*CAP1/CAP1$^{C446*}$* and SC5314 backgrounds. NAT-resistant transformants were PCR screened for correct integration of the *FLIP-NAT* construct at the *MDR1* locus using primer pairs 1716 + 1045 (left of integration) and 1636 + 2001 (right of integration).

**(vii) *MDR1* homozygous deletion.** Validated Δ*mdr1/MDR1* transformants were inoculated in YNB + BSA from frozen stocks and cultured 48 h at 220 rpm, 30°C to promote FLIP-mediated excision of *NAT*. Cultures were diluted and ~100 cells were plated on YPAD agar, then incubated for 24 h, 30°C. Colonies were patched to both YPAD and YPAD+150 µg/ml NAT agar. For colonies that were only able to grow on YPAD, PCR screening was performed to validate the correct excision of *FLIP-NAT* using primer pairs 1574 + 1575 (*NAT*) and 1716 + 2001 (across *MDR1*). Correct colonies were re-transformed with the *FLIP-NAT* construct with *MDR1* flanking sequence amplified with primers 1720 + 2000 from plasmid pJK863 [58]. NAT-resistant transformants were PCR screened for correct integration of the *FLIP-NAT* construct at the *MDR1* locus using primer pairs 1716 + 1045 (left of integration), 1636 + 2001 (right of integration), and 1273 + 2001 (inside *MDR1*). Transformants were validated for correct integration by whole-genome sequencing.

**(viii) *CAP1* tagging.** After a few attempts, *CAP1* C-terminal tagging with GFP under the native promoter did not exhibit sufficient fluorescent signal for imaging. Therefore, the *TAR-TetO-NAT* constitutive promoter replacement was transformed into the SC5314 background and PCR verified as above. After excision of *NAT,* the C-terminal *GFP-NAT* construct was PCR amplified from plasmid pMG2120 [59] using the primer pairs 2018 + 2019 (*CAP1-GFP-NAT*), or primer pairs carrying point mutation 2062 + 2041 (*CAP1$^{C446*}$-GFP-NAT*) and 2083 + 2041 (*CAP1$^{E448*}$-GFP-NAT*), and transformed into the *TetO-CAP1* background. NAT-resistant colonies were PCR screened for correct integration of the *GFP-NAT* construct at the C-terminal end of the *CAP1* locus using primer pairs 1179 + 1927 (left integration) and 1636 + 1671 (right integration). The left flank PCR confirmed that the *TetO* and *GFP-NAT* were on the same allele. Drug susceptibility was tested for all tagged strains to validate the proper function of Cap1 with or without truncation.

For tagging wild-type *CAP1* in the *CAP1/CAP1$^{C446*}$* background, the *TAR-TetO-NAT* promoter replacement was transformed into the P10-S1*CAP1/CAP1$^{C446*}$* background. After excision of *NAT,* Sanger sequencing was performed to verify that the *TetO* was on the wild-type *CAP1* allele. Then the C-terminal *RFP-NAT* construct was PCR amplified from plasmid

pMG2261 [59] using primer pair 2075 + 2019 (*CAP1-RFP-NAT*), and transformed into the *TetO-CAP1* background. NAT-resistant colonies were PCR screened for correct integration of the *RFP-NAT* construct at the C-terminal end of the *CAP1* locus using primer pairs 1179 + 2080 (left integration) and 1636 + 1671 (right integration). Drug susceptibility was tested for all tagged strains to validate the proper function of Cap1 with or without truncation.

**(ix) Wild-type control, *CAP1-NA*T.** To construct the wild-type control for murine infection, *CAP1-NAT* was PCR amplified from plasmid pAS3148 using primer pair (1670 + 1671) and transformed into the SC5314 background. NAT-resistant transformants were PCR screened for correct integration of the constructs at the *CAP1* locus using primers 1670 + 1575 (left of integration) and 1636 + 1671 (right of integration). Transformants were validated for correct integration and no off-target *CAP1* mutation by whole-genome sequencing.

### Identification of Q49* and Q298* on the same *CAP1* allele

To determine if Q49* and Q298* identified in both MEC218 and MEC219 occurred on the same allele of *CAP1*, the *CAP1* region was amplified together with upstream (primers 1674 + 1675) and downstream (primers 1676 + 1677) homology. These PCR amplified fragments and NAT (primers 1574 + 1575) were fused into the pUC19 backbone (primers 1578 + 1579) using NEBuilder HiFi assembly kit following manufacturer instructions and transformed into *E. coli*. Ampicillin-resistant bacterial transformants were screened by PCR for correct assembly using primers 1352 + 1353 and saved in glycerol stock. The *CAP1* gene of correct transformants was sequenced using Sanger sequencing to determine the co-occurrence of these two nonsense mutations.

### Microdilution MIC and SMG assay

Antifungal drug susceptibilities were determined using a microwell broth dilution assay [60]. Strains were inoculated in 2% dextrose YPAD from frozen stocks and incubated 16 h at 30°C, 220 rpm. Cultures were standardized to an $OD_{600}$ of 0.01, then diluted 1:10 into 1% dextrose YPAD media containing either 2-fold serial dilutions of the drug or a no-drug control. Drug concentrations ranged from 0.5–256 µg/ml for fluconazole, 0.008–0.25 µg/ml for itraconazole, 0.008–0.25 µg/ml for miconazole, and 0.004–2 µg/ml for micafungin. Triplicates of each strain were set up in flat-bottom 96-well plates and incubated in a humidified chamber at 30°C. At the 24 h and 48 h time points, cells were resuspended, and $OD_{600}$ was read using a BioTek Epoch2 microplate spectrophotometer. The $MIC_{50}$ of each strain was determined to be the drug concentration at which ≥50% of growth was inhibited when compared with the no-drug control at 24 h post-inoculation. The SMG was measured as the average growth ($OD_{600}$) above the $MIC_{50}$ when standardized to the no-drug control at 48 h post-inoculation. To measure viability post-FLC MIC assay, 5 µl of cells were removed from each well of the 48 h MIC plate and spotted on YPAD agar without drugs. Plates were incubated 16 h at 30°C, then imaged using a BioRad GelDoc XR+ imaging system.

### Growth curve assay

Strains were inoculated in 2% dextrose YPAD from frozen stocks and incubated 16 h at 30°C, 220 rpm. Cultures were diluted in fresh 1% dextrose YPAD to a final $OD_{600}$ of 0.01, then diluted 1:10 into 1% dextrose YPAD media supplemented with drug or a no-drug control. Cells were incubated in a BioTek Epoch2 microplate spectrophotometer at 30°C with continuous double orbital shaking (237 rpm) for 48 h, with $OD_{600}$ measurements taken every 15 min. Each isolate was performed in triplicate.

### Checkerboard assay for fluconazole and 4NQO

Strains were inoculated in 2% dextrose YPAD from frozen stocks and incubated 16 h at 30°C, 220 rpm. Cultures were diluted in fresh 1% dextrose YPAD to a final $OD_{600}$ of 0.01, then diluted 1:10 into 1% dextrose YPAD media supplemented

with drug or a no-drug control. FLC or 4NQO were 2-fold serially diluted. FLC (x-axis) ranged from 0.25−256 µg/ml and 4NQO (y-axis) ranged from 0.008–0.5 µg/ml. Cells were incubated in a BioTek Epoch2 microplate spectrophotometer at 30°C with continuous double orbital shaking (237 rpm) for 48 h, with $OD_{600}$ measurements taken every 15 min. Each isolate was performed in triplicate.

## Relative fitness assay

Isolates were inoculated in YPAD from frozen stocks and incubated for 16 h at 30°C and 220 rpm. Cultures were diluted in fresh 1% dextrose YPAD [61,62] to a final optical density ($OD_{600}$) of 0.01. Normalized cultures from two different isolates were then combined at 1:9 (sample of interest: WT-control), and 20 µl of this combined culture was added to a 96-well plate containing 180 µl of 1% dextrose YPAD supplemented with or without FLC (initial $OD_{600}$, N0 = 0.001). Cells were incubated at 30°C in a BioTek Epoch2 microplate spectrophotometer with double orbital shaking (237 rpm). $OD_{600}$ readings were taken every 15 min for 48 h to monitor cell growth, as well as at the endpoint. A volume of 20 µl culture was removed from one of the triplicates for flow cytometry at 24 h. Cultures were diluted in PBS and 10,000 singlets were gated and analysed at each time point using a Cytek Aurora flow cytometer R0021. After 16 h, populations reached the stationary phase, and the $OD_{600}$ was ~1.3. We estimated generations using the following equation: generations = [log10(Nt/N0)]/0.3, where Nt is the $OD_{600}$ at the 16 h time point, and N0 is the initial $OD_{600}$. The proportion of the sample of interest was indicated by the proportion of cells with fluorescence out of the total population, and the proportion of WT-control cells was indicated by the proportion of non-fluorescent cells out of the total population. All competitive assays were conducted in three independent replicates. Relative fitness was estimated using natural log regression analysis of the proportions of the sample of interest and the WT-control against the generations as follows: ln (proportion of sample of interest/ proportion of WT-control)/ generations.

## Filamentation

Strains were struck on YPAD agar plates from frozen stocks and incubated at 30°C for 24 h. Recovered cells were inoculated in 2% dextrose YPAD and incubated 16 h at 30°C, 220 rpm. Cultures were diluted 1:100 in RPMI+10% FBS, then cultured for 4 h at 37°C, 220 rpm. Cells were harvested, washed once with PBS, and resuspended in PBS for microscopy. Images were captured using an Olympus IX83 microscope and analyzed using ImageJ version v1.54d.

## Microscopy

**CAP1 localization.** Strains were struck on YPAD agar plates from frozen stocks and incubated at 30°C for 24 h. Recovered cells were inoculated in 2% dextrose YPAD and incubated 16 h at 30°C, 220 rpm. Cultures were diluted 1:100 in fresh YPAD, then cultured for 4 h at 30°C, 220 rpm. Cells were harvested, washed once with PBS, and resuspended in PBS for microscopy. Images were captured using an Olympus IX83 microscope and analyzed using ImageJ version v1.54d.

## Total ROS detection

Cells used for ROS staining were prepared according to the MIC method described above. At the 24 h time point, cells were resuspended and $OD_{600}$ was read using a BioTek Epoch2 microplate spectrophotometer. Cultures were harvested by centrifugation, and pooled technical replicates were washed once with PBS. Following washing, cells were resuspended in PBS supplemented with 5 mM ROS detection reagent at 1:1000 dilution (Enzo). Flow cytometry was performed for stained cells with unstained cells as the control. 10,000 singlets were gated and analyzed using a Cytek Aurora R0021 flow cytometer. Data were analyzed using FlowJo (https://www.flowjo.com/solutions/flowjo/downloads) (v10.10.0).

## Flow cytometry for GFP

Strains were struck on YPAD agar plates from frozen stocks and incubated at 30°C for 24 h. Recovered cells were inoculated in 2% dextrose YPAD and incubated 16 h at 30°C, 220 rpm. Cells were harvested, washed twice with PBS, and resuspended in PBS. Flow cytometry was performed for GFP positive cells with background strain (GFP negative) cells as the control. 10,000 singlets were gated and analyzed using a Cytek Aurora R0021 flow cytometer. Data were analyzed using FlowJo (https://www.flowjo.com/solutions/flowjo/downloads)(v10.10.0).

## Propidium iodide viability staining

Cells used for ROS staining were prepared according to the MIC method described above. Following the end of the protocol (48 h time point), cultures were harvested by centrifugation, and pooled technical replicates were washed once with PBS. Following washing, cells were resuspended in PBS supplemented with 20 μg/ml propidium iodide. Cells were stained for 10 min at 37°C, then analyzed by flow cytometry. 10,000 singlets were gated and analyzed using a Cytek Aurora R0021 flow cytometer. Data were analyzed using FlowJo (https://www.flowjo.com/solutions/flowjo/downloads) (v10.10.0).

## ROS inhibitor

The ROS inhibitor (N-acetyl-L-cysteine) from the ROS detection kit was added into MIC assay at final concentration 5 mM at 0 h time point. The $OD_{600}$ was read at both 24 h and 48 h for determining MIC and SMG. At 48 h, cells were prepared as described above for viability staining.

## RNA sequencing

**(i) RNA extraction.** Strains were struck on YPAD agar from frozen stocks and incubated at 30°C for 24 h. Three single colonies for each strain were inoculated in 2% dextrose YPAD media (50 ml) and incubated for 16 h at 30°C, 220 rpm. Cultures were diluted 1:100 into fresh 2% dextrose YPAD media (50 ml) containing no drug and YPAD supplemented with 2 μg/ml FLC, then incubated 5 h at 30°C, 220 rpm to an $OD_{600}$ of 0.5. Cells were harvested by centrifugation and snap-frozen in liquid nitrogen. RNA was prepared according to manufacturer's instructions for the Qiagen RNeasy Mini kit (Qiagen, US) using the mechanical disruption method. Removal of DNA was performed with a 1 h on-column DNase digestion at room temperature (Qiagen RNase-free DNase set, US).

**(ii) RNA sequencing.** Library preparation was performed by SeqCenter (Pittsburgh, PA) using Illumina's Stranded mRNA preparation and 10 bp unique dual indices (UDI). Sequencing was done on a NovaSeq X Plus, producing 150 bp paired-end reads. Demultiplexing, quality control, and adapter trimming were performed with bcl-convert (v4.1.5) (BCL Convert).

**(iii) RNA sequencing data analysis.** *C. albicans* transcriptome (SC5314_version_A21-s02-m09-r10_orf_coding, downloaded from http://www.candidagenome.org/download/sequence/C_albicans_SC5314/Assembly21/current/?C=S;O=A on 2023/08/17) was indexed using salmon (v1.10.2) [63]. All samples were quasi-mapped to transcriptome index using salmon, resulting in quantification of reads mapped to each transcript. The output quantification files were imported into R (v4.1.2) using tximport (v1.22.0) [64], and DESeq2 (1.34.0) [65] was used to model gene expression. PCA analysis was performed using DESeq2 and used to identify any outliers amongst the replicates. The DESeq2 'contrast' wrapper was then used to estimate log2 fold changes for each mutant relative to the wild-type control in YPAD and YPAD+2 μg/ml FLC conditions and identify differentially expressed genes. The threshold for differentially expressed genes was an absolute value log2 fold change ≥ 1 and adjusted p-value < 0.05. Differentially expressed genes in *CAP1/CAP$^{C446*}$* with or without FLC exposure are listed in S4 Data and S7 Data.

## Gene ontology and enrichment analysis

Gene ontology (GO) enrichment analyses were performed using two complementary approaches. For genes downregulated in the *CAP1*/*CAP1^{C446\*}* mutant compared to *CAP1*/*CAP1* under YPAD conditions, GO slim mapping was conducted using the *Candida* Genome Database (http://www.candidagenome.org/), with results provided in S5 Data. For Cap1 target genes showing differential expression between *CAP1*/*CAP1^{C446\*}* and *CAP1*/*CAP1* strains, Process Ontology enrichment analysis was performed using FungiDB (https://fungidb.org/fungidb/app). This analysis was applied to two gene sets: (1) genes upregulated in YPAD (results in S6 Data) and (2) genes downregulated in YPAD+2 µg/ml FLC (results in S8 Data).

## Reverse transcriptase qPCR

cDNA was prepared using the SuperScript II Reverse Transcriptase (Fisher Scientific) according to the manufacturer's instructions with oligo dT primers and 100 ng of RNA. cDNA was then diluted 1:10 with nuclease-free water for qPCR measurement. Realtime qPCR was conducted using the PowerUp SYBR Green Master Mix (Applied Biosystems) according to the manufacturer's instructions to measure cDNA. Using CFX Connect Real-Time PCR Detection System and Bio-Rad CFX Maestro software to determine Cq values, expression was calculated as the amount of cDNA from the gene of interest (*MDR1* or *CAP1*) relative to the amount of *TEF1* cDNA in the same sample. All primers used in this study are listed in S1 Data.

## Illumina whole-genome sequencing

Genomic DNA was isolated using a phenol-chloroform extraction as described previously [43]. Libraries were prepared using the Illumina DNA Prep kit and IDT 10 bp UDI indices, and sequenced on an Illumina NextSeq 2000, producing 2x151bp reads. Demultiplexing, quality control, and adapter trimming were performed with bcl-convert (https://support.illumina.com/sequencing/sequencing_software/bcl-convert.html) (v3.9.3). Adapter and quality trimming were performed with BBDuk (BBTools v38.94) [66]. Trimmed reads were aligned to the *C. albicans* reference genome (SC5314_version_A21-s02-m09-r08) using BWA-MEM (v0.7.17) with default parameters [67,68]. Aligned reads were sorted, duplicate reads were marked, and the resulting BAM file was indexed with Samtools (v1.10) [69]. Quality of trimmed FASTQ and BAM files was assessed for all strains with FastQC (v0.11.7), Qualimap (v2.2.2-dev) and MultiQC (v1.16) [70–72].

## Variant Calling and SNP Analysis

   (i) *In vitro* evolved isolates. *De novo* variant calling was performed using GATK (v.4.1.2) as previously described [39,73]. Briefly, variant calling was performed with Mutect2, assigning each progenitor as 'normal' and the evolved isolates as 'tumor'. After filtering, merged VCF files were created for each progenitor group, followed by additional hard filtering. Variants were annotated with SnpEff (v.5.0e; database built from SC5314 v.A21-s02-m09-r08, with alternate yeast nuclear codon table [74]).
   (ii)  Clinical and environmental isolates. *C. albicans* whole-genome sequencing data for 101 bloodstream isolates (BioProject PRJNA1068683) and 199 publicly available data sets (BioProjects PRJNA193498 and PRJNA432884) were processed as described above [40,41,44]. SNPs and small indels were called using HaplotypeCaller (GATK v4.4.0) [73]. SNPs were filtered on the parameters $QD < 2.00$, $QUAL < 30.0$, $SOR > 3.0$, $FS > 60.0$, $MQ < 40.0$, $MQRankSum < -12.5$, and $ReadPosRankSum < -8.0$. Indels were filtered on the parameters $QD < 2.0$, $QUAL < 30.0$, $FS > 200.0$, and $ReadPosRankSum < -20.0$. Bcftools (v1.17) was used to calculate variant allele frequency (VAF) per sample, filter for heterozygous variants with VAF between 0.15 and 0.85 and homozygous variants with $VAF > 0.98$, and to exclude known repetitive regions as annotated in the SC5314 A21-s02-m09-r08 GFF (rRNA, repeat_region, retrotransposon) and telomere-proximal regions, defined here as extending from each chromosome end to the first non-repetitive genome feature [67,69,75]. Filtered variants were annotated using SnpEff [74]. *CAP1* variants were compiled into S2 Data using GATK VariantsToTable and bcftools query.

## Visualization of whole-genome sequencing data

Chromosomal copy number changes were visualized using the Yeast Mapping Analysis Pipeline (YMAP v1.0) [76]. Aligned BAM files were uploaded to YMAP and read depth was determined and plotted as a function of chromosome position using the reference genome *C. albicans* SC5314 (A21-s02-m09-r08). Read depth was corrected for GC-content and chromosome-end bias. WGS data were plotted as the log2 ratio and converted to chromosome copy number (y-axis, 1–4 copies) as a function of chromosome position (x-axis, Chr1-ChrR). The baseline chromosome copy number (ploidy) was determined by flow cytometry (S1 Data). Haplotypes are indicated by color: gray is heterozygous (AB), magenta is homozygous BB, and cyan is homozygous AA.

## Murine model

All mouse experiments were conducted at the Lundquist Institute for Biomedical Innovation in Torrance, CA, with approval from the institution's Animal Care and Use Committee. Male, 6- week-old CD-1 mice were obtained from Envigo. The *C. albicans* strains were cultured in YPD medium at 30°C for 16 hours. Yeast cells were washed and resuspended in phosphate-buffered saline (PBS) and counted using a hemocytometer. Mice were injected intravenously through the lateral tail vein with 1 x 10^5 yeast cells. Mice received fluconazole, 2 mg/kg/day, by oral gavage for 5 consecutive days, with the first dose administered 24 hours post-infection. Mice were monitored two times daily for 21 days, and moribund mice were humanely euthanized.

## Supporting information

**S1 Fig. Heterozygous *C446\** mutation identified during fluconazole evolution.** whole-genome sequence data for the entire population at P0, P1, P2, and P3 and seven single colonies from P1 or P10. Data are plotted as the log2 ratio and converted to chromosome copy number (Y-axis, 1–4 copies) as a function of chromosome position (X-axis, Chr1–ChrR) using YMAP [76]. Arrow indicates the position of the *CAP1* locus on Chr3. Haplotypes relative to SC5314 are indicated: white is homozygous for the reference strain, grey is heterozygous AB, magenta is homozygous B, cyan is homozygous A, and blue is trisomy AAB. **B.** A subset of whole-genome sequencing reads mapped to the *CAP1* gene (Chr3 positions 480,330–480,370) for the wild-type progenitor (top) and three fluconazole-evolved single colonies from passage 10 (P10). Reads visualized using IGV [77], where gray shading indicates exact alignment to the reference genome. The T-to-A mutation in the three single colonies has an allele frequency of ~0.5 (red = reference base; green = mutant base). The mutation changes a cysteine codon (TGT) to a stop codon (TGA) at amino acid 446.
(TIF)

**S2 Fig. A.** Multiple sequence alignment of *S. cerevisiae* Yap1, *C. albicans* Cap1-A and Cap1-B from the diploid reference strain (SC5314), and Cap1 from clinical isolate MEC079. Similarity highlighted with gray. Sequence conservation relative to *S. cerevisiae* Yap1 is indicated with the heatmap (red is 100% conservation). Asterisks (*) indicate the amino acid position where nonsense variants were identified in *in vitro* and clinical isolates. Black boxes indicate the Cap1 Cysteine Rich Domains (CRD) at the N-terminus (N-CRD) and C-terminus (C-CRD). **B.** AlphaFold predicted protein structures of wild-type Cap1 (i), and truncated Cap1 with either C446* (ii), E448* (iii), Q49* (iv), or Q298* (v) variants. **C.** 24 h MIC (top, μg/ml) and 48 h SMG (bottom) in FLC for *CAP1* heterozygous overexpressed strain (*CAP1/CAP1-OE)* and *CAP1* heterozygous and homozygous deletion mutants (*CAP1/cap1Δ* and *cap1Δ/Δ*) with wild-type strain (SC5314_*CAP1/CAP1*) as the control. For MIC values, each dot represents a single replicate, and each bar represents the average of three biological replicates of a single strain; SMG values are mean ± SD calculated from three biological replicates of a single strain. The data underlying this Figure can be found in https://doi.org/10.5281/zenodo.18250101.
(TIF)

**S3 Fig. C-terminal truncation of Cap1 decreases cell viability at supra-MIC concentrations of fluconazole. A&B.** Cells from the 48 h MIC assay in Fig 2B, were plated for viability on YPAD agar plates and imaged after 24 h incubation. **A.** P10-S1_$CAP1/CAP1^{C446*}$, SC5314_$CAP1/CAP1^{C446*}$ and P10-S1_$CAP1/CAP1^{C446*}::CAP1$ were tested with wild-type strain SC5314 ($CAP1/CAP1$) as the control. White lines indicate the FLC MIC from Fig 2B. **B.** MEC079_$CAP1/CAP1^{E448*}$, SC5314_$CAP1/CAP1^{E448*}$and MEC079_$CAP1/CAP1^{E448*}::CAP1$ were tested with wild-type strain (SC5314_$CAP1/CAP1$) as the control. White lines indicate FLC MIC from Fig 2B. **C&D.** Cell viability post-FLC MIC assay of (**C**) *in vitro* evolved mutant P10-S1_ $CAP1/CAP1^{E446*}$ and its mutated allele replaced strain (P10-S1_ $CAP1/CAP1^{E446*}::CAP1$); (**D**) MEC079 _$CAP1/CAP1^{E448*}$ and its mutated allele replaced strain (MEC079 _$CAP1/CAP1^{E448*}::CAP1$) from Fig 2B and S3 Data. Cell viability was determined by propidium iodide staining post 48 h MIC assay, together with all strains and wild-type control from Fig 2C (methods). Data were assessed for normality with a Shapiro–Wilk test, and significant differences between the $CAP1$ mutant and its mutated-allele replaced strain across different FLC concentrations were calculated using two-way ANOVA with Šídák's multiple comparisons test (two-sided); * $P<0.05$, ** $P<0.01$, ns $P>0.05$; the exact $P$ values are **0.0015 and 0.0094, *0.0413 and 0.0119 for all indicated comparisons. Values are mean±SD calculated from three biological replicates. The data underlying this Figure can be found in S3_Data. **E.** Growth rate (area under the curve, AUC, Y-axis) of $CAP1$ mutants with C-terminal truncation in the presence of different concentrations of $H_2O_2$ (0−10 mM, X-axis) and 4NQO (0–0.3 μg/ml, X-axis) with wild-type as the control. Data were assessed for normality with a Shapiro–Wilk test, and significant differences between $CAP1$ mutants and wild-type control across different 4NQO concentrations were calculated using two-way ANOVA with Dunnett's multiple comparisons test (two-sided); *$P<0.05$, ** $P<0.01$; the exact $P$ values are *0.0399, **0.0030 for all indicated comparisons. The data underlying this Figure can be found in https://doi.org/10.5281/zenodo.18250101. **C-E:** Values are mean±SD calculated from three biological replicates. (TIF)

**S4 Fig. High level of intracellular ROS was detected in $CAP1$ C-terminal truncated mutant and causes decreased cell viability. A.** Checkerboard growth curve assay (left) and cell viability (right) of $CAP1/CAP1^{E448*}$ (SC5314 genetic background) in the presence of increasing concentrations of FLC (X-axis, 0–256 μg/ml, 2-fold dilutions) and/or increasing concentrations of 4NQO (Y-axis, 0–0.5 μg/ml, 2-fold dilutions). Growth rate was estimated with the area under the curve (AUC heatmap) of the 48 h growth curve. Cell viability: cells from growth curve were plated on YPAD agar and imaged after 24 h incubation. **B.** Intracellular ROS level of $CAP1/CAP1^{E448*}$ at different concentrations of FLC (FLC0-FLC16, μg/ml). Intracellular ROS was determined by ROS fluorescent detection kit combined with flow cytometry (methods). Left: Histogram of ROS fluorescent intensity from one representative biological replicate; Right: Median fluorescence intensity (MFI) of ROS and values are mean±SD calculated from three biological replicates. Comparison was between FLC-exposed cells (FLC 0.5-FLC 16, μg/ml) and no-drug control (FLC 0) for each strain. Dotted lines indicate the highest level of intracellular ROS in the wild-type strain from Fig 3C. Black box indicates the MIC of test strains. **C.** The proportion of viable cells for wild-type ($CAP1/CAP1$) and $CAP1/CAP1^{E448*}$ post-FLC MIC assay at different concentrations of FLC (FLC 0-FLC 16) with or without ROS inhibitor (ROSi, S3_Data). Cell viability was determined by propidium iodide staining (methods). Statistical comparisons were against the FLC-only treated wild-type cells across different concentrations of FLC. The data underlying this Figure can be found in S3_Data. **B&C**: Data were assessed for normality with a Shapiro–Wilk test, and significant differences (**B**) one-way ANOVA with Dunnett's multiple comparisons test and (**C**) two-way ANOVA with Dunnett's multiple comparisons test (two-sided); ****$P<0.0001$, * $P<0.05$, ns $P>0.05$; the exact $P$ value is * 0.0226. **D.** 24 h MIC (left, μg/ml) and 48 h SMG (right) in FLC with or without ROSi treatment for $CAP1/CAP1^{E448*}$ with the wild-type strain ($CAP1/CAP1$) as the control. For MIC values, each dot represents a single replicate, and each bar represents the average of three biological replicates of a single strain; SMG values are mean±SD calculated from three biological replicates of a single strain. The data underlying this Figure (A, B, and D) can be found in https://doi.org/10.5281/zenodo.18250101. (TIF)

**S5 Fig. RNA abundance changes across *CAP1*-related transcription factors of azole resistance and filamentation and *GPX3* in *CAP1* C-terminal truncated mutant. A&B:** RNA abundance of *CAP1*-related transcription factors (**A**) and *GPX3* (**B**) in wild-type strain (*CAP1/CAP1*) and *CAP1/CAP1^{C446*}*. RNA reads were normalised to transcript length and total RNA reads. Values are mean ± SD calculated from three biological replicates. Each dot represents a single replicate. **C**. Quantification of the yeast (<6 μm), pseudohyphae (15–36 μm), and hyphae (>36 μm) for *CAP1/CAP1^{C}446*}* and *CAP1/CAP1^{E448*}* with wild-type as the control. At least 100 cells were counted for each strain, and three biological replicates were performed. Values are mean ± SD calculated from three biological replicates. Statistical significance for filamentation was compared to *CAP1/CAP1* and assessed using two-way ANOVA with Dunnett's multiple comparisons test, ****$P < 0.0001$. The data underlying this Figure can be found in https://doi.org/10.5281/zenodo.18250101.
(TIF)

**S6 Fig. Transcription activation of both wild-type and mutated *CAP1*. A.** Gene Ontology (GO) terms for upregulated Cap1 targets in the *CAP1/CAP1^{C446*}* mutant compared to *CAP1/CAP1* in YPAD (S6_Data). The data underlying this Figure can be found in S6_Data. **B.** Gene Ontology (GO) terms for downregulated Cap1 targets in the *CAP1/CAP1^{C446*}* mutant compared to *CAP1/CAP1* in FLC (S8_Data). The data underlying this Figure can be found in S8 Data. **A&B**: X-axis: fold enrichment relative to the wild-type reference strain. Circle size indicates the gene counts of each term. The intensity of red indicates the adjusted *P*-values calculated by the Benjamini-Hochberg procedure. **C.** Expression changes (log2 fold change) of Cap1 target genes that are involved in oxidative stress response and drug transport in FLC relative to YPAD within wild-type (*CAP1/CAP1*) (left) and *CAP1/CAP1^{C446*}* mutant (right) background. Gene ontology terms were adopted from [15]. The data underlying this Figure can be found in https://doi.org/10.5281/zenodo.18250101. **D.** Subcellular localization of Cap1 with E448* (*TetO-CAP1^{E448*}-GFP/CAP1*), scale bar is 5 μm. Hoechst (light blue) was applied, indicating nuclear co-localization (Methods).
(TIF)

**S7 Fig. C-terminal truncated *CAP1* turns multiple azoles into fungicides. A.** 24 h MIC (μg/ml) (Top) and 48 h SMG in micafungin (MFG), itraconazole (ITC), miconazole (MICO), and voriconazole (VOC) for *CAP1/CAP1^{C446*}*, *CAP1/CAP1^{E448*}*, and *MDR1* overexpression strains *MDR1-OE_1* and *MDR1-OE_2,* with wild-type background strain (SC5314, *CAP1/CAP1*) as the control. **B**. Cell viability post-FLC MIC assay. The proportion of viable cells for wild-type (*CAP1/CAP1*) and *CAP1/CAP1^{E448*}* post 48 h MIC assay at different concentrations of ITC, MICO, and VOC (0–0.25 μg/ml). Cell viability was determined by propidium iodide staining (methods). Comparison was between *CAP1/CAP1* and *CAP1/CAP1^{E448*}* across different concentrations of drugs. Data were assessed for normality with a Shapiro–Wilk test, and significant differences using two-way ANOVA with Šídák's multiple comparisons test (two-sided); ** $P < 0.01$,*** $P < 0.001$,**** $P < 0.0001$, ns $P > 0.05$; the exact P values are * 0.0144, ** 0.0012 and 0.0039, ***0.0010, and **** < 0.0001. **C.** 48 h SMG in cerulenin (CER) for strains *CAP1/CAP1^{C446*}*, *CAP1/CAP1^{E448*}* and *MDR1* overexpression strains *MDR1-OE_1* and *MDR1-OE_2,* with wild-type background strain (SC5314, *CAP1/CAP1*) as the control. **D.** 48 h SMG in FLC for FLC-evolved mutant P10-S1 (*CAP1/CAP1^{C446*}*) and *mdr1Δ/Δ_1* and _2 in P10_S1 background, with wild-type strain (*CAP1/CAP1*) and *mdr1Δ/Δ_1&_2* in wild type background as the controls. Two independent transformants were included for *MDR1* deletion mutants as the control. **A&B&D:** values are mean ± SD calculated from three biological replicates of a single strain and each dot represents a single replicate. The data underlying this Figure can be found in https://doi.org/10.5281/zenodo.18250101.
(TIF)

**S1 Data. Strains and primers used in this study.**
(XLSX)

**S2 Data. Summary of *CAP1* sequence variants from 300 *C. albicans* clinical isolates.**
(XLSX)

**S3 Data. Cell viability of *CAP1* nonsense mutants and mutated-allele replaced strains compared to wild-type in fluconazole (FLC) with or without the ROS inhibitor (ROSi).**
(XLSX)

**S4 Data. Differentially expressed genes in *CAP1*/*CAP1*C446* compared to *CAP1*/*CAP1* in YPAD.**
(XLSX)

**S5 Data. Gene ontology terms for downregulated genes in *CAP1*/*CAP1*C446* compared to *CAP1*/*CAP1* in YPAD.**
(XLSX)

**S6 Data. Gene ontology terms for 51 upregulated Cap1 targets in *CAP1*/*CAP1*C446* compared to *CAP1*/*CAP1* in YPAD.**
(XLSX)

**S7 Data. Differentially expressed genes in *CAP1*/*CAP1*C446* compared to *CAP1*/*CAP1* in 2 µg/ml fluconazole.**
(XLSX)

**S8 Data. Gene ontology terms for 23 downregulated Cap1 targets in the *CAP1*/*CAP1*C446* mutant compared to *CAP1*/*CAP1* in 2 µg/ml fluconazole.**
(XLSX)

## Acknowledgments

We are grateful to Berman lab, Cowen lab and Köhler lab for providing plasmids for strain engineering: pMG2,120, pMG2,261, pLC605, and pJK863. We are grateful to Laura S. Burrack, Petra Vande Zande, Jerzy Twarowski, and Mark McClellan for helpful discussions and feedback on the manuscript.

## Author contributions

**Conceptualization:** Xin Zhou, Anna Selmecki.

**Data curation:** Xin Zhou, Audrey Hilk.

**Formal analysis:** Xin Zhou, Audrey Hilk, Norma V Solis, Nancy Scott.

**Funding acquisition:** Anna Selmecki.

**Investigation:** Xin Zhou, Audrey Hilk, Norma V Solis.

**Methodology:** Xin Zhou, Norma V Solis, Nancy Scott, Christopher Zajac.

**Project administration:** Scott G Filler, Anna Selmecki.

**Resources:** Scott G Filler, Anna Selmecki.

**Software:** Xin Zhou.

**Supervision:** Scott G Filler, Anna Selmecki.

**Validation:** Xin Zhou, Christopher Zajac, Anna Selmecki.

**Visualization:** Xin Zhou.

**Writing – original draft:** Xin Zhou, Audrey Hilk, Anna Selmecki.

**Writing – review & editing:** Xin Zhou, Anna Selmecki.

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
