## [Editor Report · Decision Letter 0]

8 Oct 2025

Dear Dr Selmecki,

Thank you for submitting your manuscript entitled "Recurrent CAP1 mutations reveal a trade-off between drug resistance and oxidative stress response in C. albicans" for consideration as a Research Article by PLOS Biology. I am currently handling your manuscript since my colleague Melissa Vazquez Hernandez is away from the office this week. Please accept my sincere apologies for the delay in getting back to you as we consulted with an academic editor about your submission.

Your manuscript has now been evaluated by the PLOS Biology editorial staff, as well as by an academic editor with relevant expertise, and I am writing to let you know that we would like to send your submission out for external peer review.

Once your full submission is complete, your paper will undergo a series of checks in preparation for peer review. After your manuscript has passed the checks it will be sent out for review. To provide the metadata for your submission, please Login to Editorial Manager (https://www.editorialmanager.com/pbiology) within two working days, i.e. by Oct 10 2025 11:59PM.

Kind regards,

Richard

Richard Hodge, PhD

rhodge@plos.org

On behalf of:

Melissa Vazquez Hernandez, Ph.D.

Associate Editor, PLOS Biology

---

## [Decision Letter · Decision Letter 1]

14 Nov 2025

Dear Anna,

Thank you for your patience while your manuscript "Recurrent CAP1 mutations reveal a trade-off between drug resistance and oxidative stress response in Candida albicans" went through peer-review at PLOS Biology. Your manuscript has now been evaluated by the PLOS Biology editors, an Academic Editor with relevant expertise, and by three independent reviewers. My sincere apologies for the really long delay on the decision.

As you will see in the reports, all reviewers are positive about the study but give several suggestions that we will require you to address as we think they will make the manuscript stronger and clearer. Reviewer 1 suggests clarifying transcriptional comparisons between WT and truncation mutants, examining Cap1 stability using existing GFP/RFP fusions, and refining interpretation and presentation of nuclear-localization microscopy. Reviewer 2 requests a control (parental mdr1Δ/Δ strain in Fig 5D), and recommends clarifying the allele-replacement experiment in Fig 2B as well as adjusting the abstract to avoid implying that many clinical isolates carry similar truncations. Reviewer 3 recommends moving certain supplemental figures into the main figures and refining interpretation of the microscopy data.

**IMPORTANT - SUBMITTING YOUR REVISION**

*Resubmission Checklist*

*Published Peer Review*

*PLOS Data Policy*

*Blot and Gel Data Policy*

Sincerely,

Melissa

Melissa Vazquez Hernandez, Ph.D.

Associate Editor

PLOS Biology

REVIEWERS' COMMENTS

Reviewer #1:

This study has identified acquired mutations in the Cap1 transcription factor that promote resistance of Candida albicans to sub-MIC concentrations of fluconazole, but intriguingly are more sensitive to supra-MIC fluconazole levels due to impaired oxidative stress responses. This provides novel insight into the relationship between antifungal drug and oxidative stress resistance, providing potential therapeutic strategies to offset drug resistance.

In this study heterozygous mutations in CAP1, resulting in the C-terminal truncation and removal of the nuclear export sequence from this transcription factor, were found following in vitro fluconazole evolution experiments (Cap1/Cap1C446*) and in a bloodstream isolate (Cap1/Cap1E448*). Convincing data is presented that both mutations increased the MIC of fluconazole. However, perhaps counterintuitively, despite increases in the MIC, strains expressing truncated Cap1 were more sensitive than wild-type cells to fluconazole concentrations above the MIC. Subsequent experiments revealed that at supra-MIC levels of fluconazole, the C-terminal truncated mutants accumulated higher levels of ROS than wild-type cells. This likely underpins the increased sensitivity of the C-terminal truncated mutants to fluconazole, as this could be reversed by addition of the antioxidant N-acetylcysteine. Gene expression analysis revealed that C-terminal truncation of Cap1 resulted in the activation of many Cap1 target genes including those involved in the oxidative stress response and drug transport, although this was impacted upon fluconazole treatment where several Cap1-dependent genes were down-regulated compared to wild-type cells. Subsequent mutational analysis indicated that upregulation of MDR1 contributed to the increased MIC to fluconazole (and cerulenin) in the Cap1 C-terminal truncated mutants. A final key finding was that the nonsense mutation at C446 conferred resistance to fluconazole in a mouse model of systemic disease.

Below are a number of suggestions to potentially further improve the manuscript:

Considering the expression of Cap1 target genes, I'm wondering if different comparisons may give extra information regarding the transcriptional responses to fluconazole in Wt and Cap1/Cap1C446 cells. For example, it would appear that OSR genes are down-regulated in Cap1/Cap1C446 cells compared to wild-type cells following fluconazole treatment. However, could this be due to a significant increase in expression in Wt cells over the high but fixed 'basal' level of expression in Cap1/Cap1C446 cells? So rather than being downregulated the expression of OSR genes in the Cap1/Cap1C446 cells doesn't change with fluconazole, whereas a significant induction is seen in Wt cells? Calculating the relative fold induction in Wt -/ Wt + fluconazole, and in Cap1/Cap1C446 -/ Cap1/Cap1C446+ fluconazole would give information as to whether the magnitude of the response is impacted - ie is there any further induction of Cap1 target genes in Cap1/Cap1C446 cells in response to fluconazole, and how does the level of expression compare to that seen in wild-type cells?

Using the GFP fusions did you examine the impact of the C-terminal truncations on Cap1 stability? Previous mutation of the c-CRD or truncation of Cap1 results in reduced protein levels (refs 15, 17), possibly through inhibiting association with Ybp1 which is also crucial for Cap1 stability (ref 24).

The unexpected finding in this study that azoles at supra-MIC concentrations exert fungicidal effects due to increased ROS accumulation in CAP1-C-terminal truncated mutants, shares some similarities to the unexpected synergistic killing of C. albicans observed following combinations of oxidative and osmotic stress. This synergistic killing was due to elevated levels of ROS which prevented Cap1 activity (ref 22). Whilst I think it is outside the scope of this work to look at post-translational modification of the C-terminal truncated Cap1 mutants to different fluconazole concentrations - some reference to the fact that increases in intracellular ROS have previously been shown to impact on Cap1 activation could be useful.

Minor points:

Figure 1C. It is not clear what Figure 1C shows with regard to the Cap1 mutation?

Line 217. The conclusion that high concentrations of FLC are fungicidal to cells carrying a CAP1 C-terminal truncation does not hold true for the MEC079 Cap1/Cap1E448* mutant (Figure S3C). Also legend for Fig 2C needs clarifying in that only the C446* mutants were examined.

Figure S6 spelling of nuclear

Figure 4E - the CAP1-RFP fusion appears to form discrete loci (not nuclear) rather than being cytoplasmic?

Figure 6. Can the authors share their thoughts as to why the patient derived Cap1E448 mutant does not result in drug resistance in vivo in contrast to Cap1C446.

Reviewer #2:

This is a very interesting paper from the Selmecki group exploring the role of mutations in CAP1 in fluconazole resistance. They demonstrate that c-terminal truncation of Cap1 results in constitutive activation and an increase in fluconazole MIC but at the cost of increased cell death at supra-MIC levels of fluconazole or other azole drugs. I largely found the claims made here to be well supported, although there is one case where I'd like to see an additional control (below). I also found one line in the abstract to be slightly misleading. The authors mention identifying 300 additional clinical isolates and identifying 25 distinct CAP1 missense or nonsense alleles. Prior to reading the text, I took this to mean there were many additional clinical cases with similar phenotypes to the Cap1 truncation allele. Rather, there is exactly one experimentally evolved line and one clinical isolate with this genotype/phenotype and no others with truncations in similar locations. I think reordering the abstract a little would alleviate this issue for me (suggestion below). Neither of these issues significantly dampened my enthusiasm for the story presented here. The somewhat paradoxical cost of the increased resistance is quite interesting and the authors performed a convincing set of experiments to tie this phenotype to redox stress. I think this will be of broad interest to the community working on antifungal drug resistance.

Major comments:

I am a little confused by the mechanism of Cap1 truncation resistance as depicted in Figure 2B. The authors propose that this is the result of a c-terminal truncation that results in increased activity of Cap1 by eliminating the nuclear export signal and causing constitutive activation of Cap1's client genes. I would predict this mechanism to result in a dominant phenotype for the Cap1 truncation alleles, especially given that it is conferring resistance as a heterozygous allele. However, Figure 2B shows complementation of this allele via addition of another copy of Cap1 that returns it to a wildtype phenotype. I think perhaps this experiment is actually an allele replacement of the truncation allele, but it is not clear from the text in the results or the genotype in the figure. If so, describing this more clearly and altering the genotype to reflect what is actually happening in this experiment would help.

Lines 16-17 in the abstract come out of order with the order told in the main text. I would move this up to line 5. As it stands this led me to expect an expanded story including many strains with similar phenotypes, although that case is not made here.

Figure 5D would benefit from a parental control mdr1 double deletion. The authors use this data to make the case that mdr1 overexpression is responsible for the increase in resistance in the Cap1 truncation alleles. I suspect they are right but without seeing what the null looks like, it is hard to distinguish this hypothesis from the competing hypothesis that mdr1 nulls simply decrease resistance in all contexts.

Minor Comments:

There are a number of spelling mistakes throughout the figures, some of which are documented below. I would take a thorough editing pass through the figures, checking for spelling.

Fig 1C: I wish there was a more visual way of depicting that these were CAP1/CAP1C446* hets in the figure here. Perhaps an inset on the CAP1 locus? The figure legend here also does not explain what a white background means in the figure (I'm assuming no diversity to score heterozygosity?).

Fig 2C: Viability is spelled incorrectly in the title

Fig 2C: I think this assay would benefit from another set of experiments at 1 ug/mL that would show whether viability defects begin above the MIC or below.

Line 243-244: Opening phrase of this sentence is not quite grammatically correct

Fig 3E: Viability spelled incorrectly again

Figure S3: Viability spelled incorrectly in a few ways here

Figure S6 nuclear is spelled incorrectly

Reviewer #3: PBIOLOGY-D-25-03189R1

This manuscript characterizes the CAP1 mutations that confer azole resistance in clinical isolates of Candida albicans. Understanding mechanisms of drug resistance is important to understand how to improve management and treatment of candidiasis in the face of growing azole resistance. The work presented here is the first to characterize mutations of CAP1 in clinical isolates and their role in fitness, drug resistance and the oxidative stress response. This is a well-designed and robust study with a few minor comments to address prior to publication:

1. Figure 1C is hard to read/see colors; I think this figure would fit better in the supplemental figures while Supplemental Figure 1 should be moved to the main figures

2. Similarly, I think the fluconazole treated cells for nuclear localization microscopy should be included in the main figure. I don't necessarily agree with the interpretation of the nuclear localization microscopy results. In the fluconazole treated samples (Fig S6D) wild type CAP1 does maintain diffuse cytoplasmic staining but there are a few clear puncta that localize with the nucleus unlike in the YPD samples where it is truly diffuse. However, the Cap1C446* treated with fluconazole look more like the CAP1-RFP (Figure 4E) where staining is a diffuse-puncta that is adjacent to the nucleus but does not overlap. Even in the CAP1-C446*-GFP images in figure 4E, there are bright puncta that overlap with the nuclei that are also associated with a diffuse staining adjacent to the nucleus. I agree that there are differences between the WT and mutant in YPD vs FLC, however it is not as simple as either being in the nucleus or diffuse cytoplasmic staining.

---

## [Decision Letter · Decision Letter 2]

13 Jan 2026

Dear Anna,

Thank you for your patience while we considered your revised manuscript "Recurrent CAP1 mutations reveal a trade-off between drug resistance and oxidative stress response in Candida albicans" for publication as a Research Article at PLOS Biology. This revised version of your manuscript has been evaluated by the PLOS Biology editors, the Academic Editor and the original reviewers.

Based on the reviews, we are likely to accept this manuscript for publication, provided you satisfactorily address the remaining editorial points. Please also make sure to address the following data and other policy-related requests.

1) We routinely suggest changes to titles to ensure maximum accessibility for a broad, non-specialist readership, and to ensure they reflect the contents of the paper. In this case, we would suggest a minor edit to the title, as follows. Please ensure you change both the manuscript file and the online submission system, as they need to match for final acceptance:

"Recurrent mutations in the stress regulator Cap1 reveal a trade-off between azole resistance and oxidative stress response in Candida albicans"

2) Please add the weblinks of the funding agencies in the Financial Disclosure statement in the manuscript details.

3) Please include the protocol/permit/project license number on your Ethics Statement.

4) We do not have a word limit. Please move the references is in the supplement to the main text which can provide the readers an easier access to all information, and provide the rightful merit.

5) Please note that per journal policy, the model system/species studied should be clearly stated in the abstract of your manuscript.

Please supply the numerical values either in the a supplementary file or as a permanent DOI’d deposition for the following figures:

Figure 1ABC, 2BC, 3A-F, 5C, 4C, 6BCD, 7AB, S2C, S3CDE, S4BCD, S5AB, S6ABC, S7ABCD

7) Please cite the location of the data clearly in all relevant main and supplementary Figure legends, e.g. “The data underlying this Figure can be found in S1 Data” or “The data underlying this Figure can be found in https://doi.org/10.5281/zenodo.XXXXX”

8) Please ensure that you are using best practice for statistical reporting and data presentation. These are our guidelines https://journals.plos.org/plosbiology/s/best-practices-in-research-reporting#loc-statistical-reporting and a useful resource on data presentation https://journals.plos.org/plosbiology/article?id=10.1371/journal.pbio.1002128

— If you are reporting experiments where n ≤ 5, please plot each individual data point.

9) For figures containing FACS data (Figures 3CD, 5C, S4B), please provide the FCS files and a picture showing the successive plots and gates that were applied to the FCS files to generate the figure. We ask that you please deposit this data in an open repository like Zenodo and provide the accession number/URL of the deposition in the Data Availability Statement in the online submission form.

10) Supplementary files (e.g., excel). Please ensure that all data files are uploaded as 'Supporting Information' and are invariably referred to (in the manuscript, figure legends, and the Description field when uploading your files) using the following format verbatim: S1 Data, S2 Data, etc. Multiple panels of a single or even several figures can be included as multiple sheets in one excel file that is saved using exactly the following convention: S1_Data.xlsx (using an underscore).

11) Please ensure that your Data Statement in the submission system accurately describes where your data can be found and is in final format, as it will be published as written there

12) Per journal policy, if you have generated any custom code during the course of this investigation, please make it available without restrictions. Please ensure that the code is sufficiently well documented and reusable, and that your Data Statement in the Editorial Manager submission system accurately describes where your code can be found. More information on our Code Policy, what and how to share can be found here: https://journals.plos.org/plosbiology/s/code-availability

We expect to receive your revised manuscript within two weeks.

*Published Peer Review History*

*Press*

Sincerely,

Melissa

Melissa Vazquez Hernandez, Ph.D.

Associate Editor

PLOS Biology

REVIEWERS' COMMENTS

Reviewer #1:

The authors have thoroughly addressed all of the comments raised. The revised wording of the downregulation of the OSR genes in CAP1/CAP1C446* cells fits with the thinking that this is linked to lack of a further OSR response in response to FLU (which is stimulated in Wt cells). The analysis of the stability of Cap1-GFP fusions has added new information as to the impact of the mutations at positions 446 or 448, and likely contributes to the phenotypic differences observed with these mutations. I also appreciate the improved clarification of the genetic backgrounds of the C446* and E448* mutants, the ability to easily compare the impact of each mutation in the same genetic background, and the clear explanation of why the clinical MEC079_CAP1/CAP1E448* isolate may display phenotypic differences to the engineered SC5314_ CAP1/CAP1E448* isolate.

Reviewer #2:

I enjoyed the first round of the paper and I think the revised version has addressed all of the reviewer comments. I have no remaning comments. Congratulations to the authors on a nice piece of work.

Reviewer #3:

The revision of this paper addresses my previous concerns and includes addtitional data that enhances rigor of the manuscript.

---

## [Editor Report · Decision Letter 3]

19 Jan 2026

Dear Anna,

Thank you for the submission of your revised Research Article "Recurrent mutations in the stress regulator Cap1 reveal a trade-off between azole resistance and oxidative stress response in Candida albicans" for publication in PLOS Biology. On behalf of my colleagues and the Academic Editor, Csaba Pál, I am pleased to say that we can in principle accept your manuscript for publication, provided you address any remaining formatting and reporting issues. These will be detailed in an email you should receive within 2-3 business days from our colleagues in the journal operations team; no action is required from you until then. Please note that we will not be able to formally accept your manuscript and schedule it for publication until you have completed any requested changes.

IMPORTANT: Many thanks for providing the protocol number in the Ethics Statement. However, we require that the statement is the first subheading in the Material & Methods section. Could you please move it? I have asked my colleagues to include this request alongside their own.

PRESS

Sincerely,

Melissa

Melissa Vazquez Hernandez, Ph.D., Ph.D.

Associate Editor

PLOS Biology
